# Linearly Constrained Diffusion Implicit Models

**Vivek Jayaram**[1]     **Ira Kemelmacher-Shlizerman**[1]     **Steven M. Seitz**[1]

**John Thickstun**[2]

[1]University of Washington [2]Cornell University

## Abstract

We introduce Linearly Constrained Diffusion Implicit Models (CDIM), a fast and accurate approach to solving noisy linear inverse problems using diffusion models. Traditional diffusion-based inverse methods rely on numerous projection steps to enforce measurement consistency in addition to unconditional denoising steps. CDIM achieves a 10–50× reduction in projection steps by dynamically adjusting the number and size of projection steps to align a residual measurement energy with its theoretical distribution under the forward diffusion process. This adaptive alignment preserves measurement consistency while substantially accelerating constrained inference. For noise-free linear inverse problems, CDIM exactly satisfies the measurement constraints with few projection steps, even when existing methods fail. We demonstrate CDIM's effectiveness across a range of applications, including super-resolution, denoising, inpainting, deblurring, and 3D point cloud reprojection. Code and an interactive demo can be found on our project website. [1]

## 1   Introduction

Recovering an unknown signal from noisy linear measurements is a fundamental challenge encompassing tasks like super-resolution, inpainting, and denoising. These are examples of linear inverse problems, for which we observe a partial measurements $\mathbf{y} \in \mathbb{R}^d$ that was generated through a linear operator $\mathbf{A} \in \mathbb{R}^{d \times n}$ applied to a signal $\mathbf{x} \in \mathbb{R}^n$. In the noisy case, measurements $\mathbf{y}$ are corrupted with noise sampled i.i.d. from a gaussian distribution with variance $\sigma_y^2$:

$$\mathbf{y} = \mathbf{A}\mathbf{x} + \boldsymbol{\sigma}, \quad \text{where } \boldsymbol{\sigma} \sim \mathcal{N}(0, \sigma_y^2 \mathbf{I}) \tag{1}$$

We aim to use a pretrained diffusion model [1] as a prior to solve these underdetermined inverse problems. However, adapting diffusion models to solve inverse problems has traditionally presented two key challenges. First, recovering the original signal from partial measurements is computationally expensive, requiring numerous additional network evaluations at each step to guide the diffusion towards the measurement constraint. Second, we would like to guarantee that the constraints are met to arbitrary precision in the noiseless case, a difficult task when trying to reduce overall steps.

A common approach to diffusion for inverse problems, used by methods like Diffusion Posterior Sampling (DPS) [2] and others works [3, 4], is to alternate unconditional diffusion steps with projection steps on the measurement error: $\nabla_{\mathbf{x}_t} \|\mathbf{A}\hat{\mathbf{x}}_0(\mathbf{x}_t) - \mathbf{y}\|^2$, where $\hat{\mathbf{x}}_0(\mathbf{x}_t)$ is an estimate of the posterior mean $\mathbb{E}[\mathbf{x}_0|\mathbf{x}_t]$. However, the optimal number and size of these projection steps is difficult to determine as care must be taken to ensure that (1) the steps are sufficient to converge on the measurement criteria, (2) unnecessary projection steps do not increase inference time, and (3) large

---

[1]https://grail.cs.washington.edu/projects/cdim/

39th Conference on Neural Information Processing Systems (NeurIPS 2025).

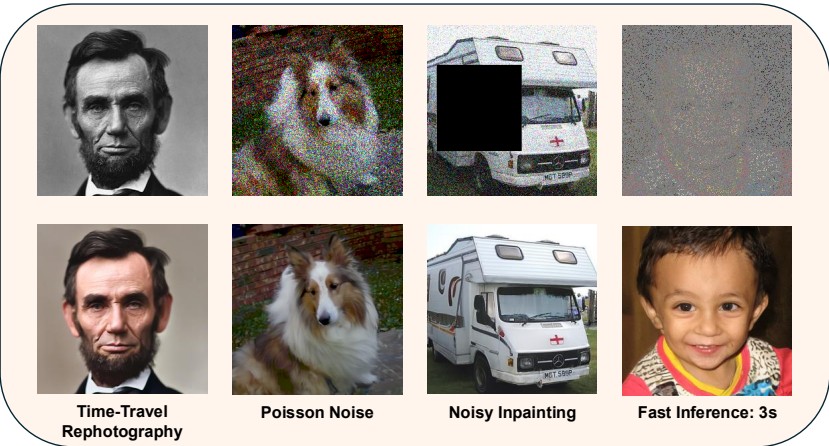

Figure 1: We show several applications of our method including image colorization, denoising, inpainting, and sparse recovery. We highlight the fact that we can handle general noise distributions, such as Poisson noise, and that our method runs in as little as 3 seconds.

step sizes do not cause divergence or pull the iterate $\mathbf{x}_t$ out of distribution. In light of these challenges, methods like DPS for many small projection steps which each require a pass through the model. Therefore, despite advances in accelerating unconditional sampling, reliance on numerous small projection steps still limits the speed of inference under constraints.

We propose linearly constrained diffusion implicit models (CDIM) to address these challenges. CDIM is a new algorithm for projection-based DDIM sampling that can generate images satisfying measurement constraints under highly accelerated sampling schedules. Our key insight is to guide the number and size of the projection steps with the known distribution of $\|\mathbf{Ax}_t - \mathbf{y}\|^2$ under the forward noising process given $\mathbf{y}$. This is a tractable chi-squared distribution, and we show that forcing $\|\mathbf{Ax}_t - \mathbf{y}\|^2$ to stay within a plausible region similarly enforces the optimization target $\|\mathbf{A\hat{x}}_0(\mathbf{x}_t) - \mathbf{y}\|^2$ to stay within a plausible region under the forward noising process. Notably the former is computable without a network evaluation, allowing us to apply strategies like binary search or line search on step size without costly overhead.

We emphasize that the speedup of CDIM **does not** simply come from adopting an accelerated unconditional DDIM schedule, but rather from a smarter projection strategy that greatly reduces the number of projection steps required for constrained sampling. This allows outputs to match partial observations under accelerated sampling *without* introducing a large number of additional network evaluations. In contrast, works such as [4] use DDIM as a sampling strategy yet still require 200 additional network evaluations for constrained sampling, showing that DDIM alone is not enough to accelerate the entire proecess. To further demonstrate this, we present qualitative examples that simply using DPS with DDIM accelerated sampling yields blurry or divergent results, even when the projection step size is tuned for optimal performance.

Furthermore, CDIM exactly recovers noiseless measurements, even for out-of-distribution inputs (impossible with DPS), while requiring 10-50x fewer projection steps. Under Gaussian measurement noise, our step size criterion naturally generalizes from the noiseless case without introducing additional network evaluations. We further extend CDIM to handle Poisson noise through a reformulation based on Pearson residuals. Empirically, CDIM achieves high-quality reconstructions in under 2.5 seconds, whereas prior methods such as DPS and MCG [3] require over 70 seconds. Across tasks including super-resolution, box inpainting, deblurring, and random inpainting, CDIM delivers comparable or superior reconstruction quality while operating an order of magnitude faster.

## 2 Related Work

Diffusion models [1] have emerged as powerful generative models, building upon early work in nonequilibrium thermodynamics [5] and implicit models [6]. Denoising Diffusion Implicit Models (DDIM) [7] was a notable work that improved the efficiency of diffusion sampling through non-

markovian sampling. This was further advanced through stochastic differential equations [8] and numerical ODE solvers like PNDM [9].

Applying diffusion models to inverse problems has been an active research area. DPS [2] was a notable method that uses alternating projection steps to guide the diffusion process. DDNM [10], DDRM [11], SNIPS [12], and PiGDM [13] use linear algebraic approaches and singular value decompositions. Techniques such as DMPS [14], FPS [15], LGD [16], DPMC [4], and MCG [17], and DAPs [18] focus on likelihood approximation for improved sampling. Guidance mechanisms have been incorporated through classifier gradients [19], data consistency enforcement [20], and low-frequency feature matching [21].

Other approaches use projection [22, 3] or optimization [23, 24, 25, 26]. DMPlug [27] backpropagates through the entire diffusion process, leading to extremely slow inference. DSG [28] uses a similar optimization update to us for enforcing consistency with the partial observation; however, it does not guarantee matching a constraint exactly, instead using a soft constraint, like DPS, to handle observational noise (see Appendix F.1). Finally, works such as Blind DPS [29] and FastEM [30] solve inverse problems when the forward operator is unknown, a more difficult problem than the setting studied in this work.

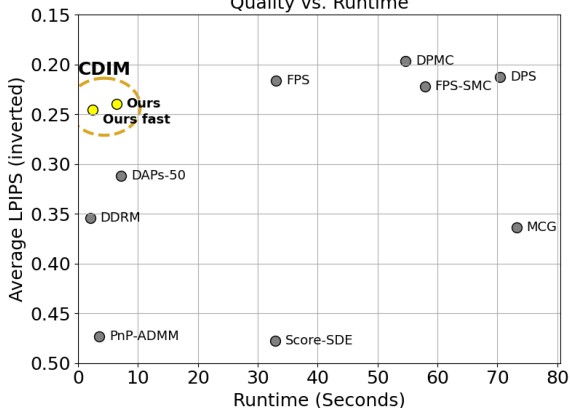

Figure 2: The family of CDIM methods (top left corner) simultaneously achieves strong generation strong quality and extremely fast inference compared to other inverse solvers. We plot the inference speed and average LPIPS image quality score (inverted) averaged across multiple inverse tasks on the FFHQ dataset. "Ours" uses $T' = 50$ denoising steps while "Ours fast" uses $T' = 25$ denoising steps

## 3 Background

We work in the context of DDPM [1], which models a data distribution $q(\mathbf{x}_0)$ by modeling a sequence $t = 1, \dots, T$ of smoothed distributions defined by

$$q(\mathbf{x}_t|\mathbf{x}_0) = \mathcal{N}(\mathbf{x}_t; \sqrt{\bar{\alpha}_t}\mathbf{x}_0, (1 - \bar{\alpha}_t)\mathbf{I}). \tag{2}$$

The degree of smoothing is controlled by a monotone decreasing noise schedule $\bar{\alpha}_t$ with $\bar{\alpha}_0 = 1$ (no noise) and $\bar{\alpha}_T = 0$ (pure Gaussian noise).[2] The idea is to model a *reverse process* $p_\theta(\mathbf{x}_{t-1}|\mathbf{x}_t)$ that that incrementally removes the noise in $\mathbf{x}_t$ such that $p_\theta(\mathbf{x}_T) = \mathcal{N}(\mathbf{x}_T; 0, \mathbf{I})$ and $p(\mathbf{x}_0)$ approximates the data distribution, where $p(\mathbf{x}_0)$ is the marginal distribution of outputs from the reverse process:

$$p_\theta(\mathbf{x}_0) = \int p_\theta(\mathbf{x}_T) \prod_{t=1}^{T} p_\theta(\mathbf{x}_{t-1}|\mathbf{x}_t) \, d\mathbf{x}_{1:T}. \tag{3}$$

Given noisy samples $\mathbf{x}_t = \sqrt{\bar{\alpha}_t}\mathbf{x}_0 + \sqrt{1 - \bar{\alpha}_t}\boldsymbol{\epsilon}$, where $\mathbf{x}_0$ is a sample from the data distribution and $\boldsymbol{\epsilon} \sim \mathcal{N}(0, \mathbf{I})$, a diffusion model $\boldsymbol{\epsilon}_\theta(\mathbf{x}_t, t)$ is trained to predict $\boldsymbol{\epsilon}$:

$$\min_\theta \mathbb{E}_{\mathbf{x}_t, \boldsymbol{\epsilon}} \left[ \|\boldsymbol{\epsilon} - \boldsymbol{\epsilon}_\theta(\mathbf{x}_t, t)\|^2 \right]. \tag{4}$$

---

[2]We define $\bar{\alpha}_t$ using the DDPM convention [1]; this corresponds to $\alpha_t$ in DDIM [7].

To parameterize the reverse process $p_\theta(\mathbf{x}_{t-1}|\mathbf{x}_t)$, DDIM [7] exploits the Tweedie formula [31] for the posterior mean of a noisy observation:

$$\mathbb{E}\left[\mathbf{x}_0|\mathbf{x}_t\right] = \frac{1}{\sqrt{\bar{\alpha}_t}}\left(\mathbf{x}_t + (1 - \bar{\alpha}_t)\nabla_{\mathbf{x}_t} \log q(\mathbf{x}_t)\right). \tag{5}$$

Using the denoising model $\boldsymbol{\epsilon}(\mathbf{x}_t, t)$ as a plug-in estimate of the score function via the relationship $\mathbb{E}\left[\boldsymbol{\epsilon}|\mathbf{x}_t\right] = -\sqrt{1 - \bar{\alpha}_t}\nabla_{\mathbf{x}_t} \log q(\mathbf{x}_t)$, we define the Tweedie estimate of the posterior mean:

$$\hat{\mathbf{x}}_0(\mathbf{x}_t) \equiv \frac{1}{\sqrt{\bar{\alpha}_t}}\left(\mathbf{x}_t - \sqrt{1 - \bar{\alpha}_t}\boldsymbol{\epsilon}_\theta(\mathbf{x}_t, t)\right) \approx \mathbb{E}\left[\mathbf{x}_0|\mathbf{x}_t\right]. \tag{6}$$

Throughout this paper, we keep the formula for the Tweedie estimate in functional form $\hat{\mathbf{x}}_0(\mathbf{x}_t)$ to make it clear that $\hat{\mathbf{x}}_0$ is a function of our current iterate $\mathbf{x}_t$. We can then use this estimator to define a DDIM forward process $\mathbf{x}_{t-1} = f_\theta(\mathbf{x}_t)$ defined by

$$\mathbf{x}_{t-1} = f_\theta(\mathbf{x}_t) = \sqrt{\bar{\alpha}_{t-1}} \cdot \hat{\mathbf{x}}_0(\mathbf{x}_t) + \sqrt{1 - \bar{\alpha}_{t-1}}\left(\frac{\mathbf{x}_t - \sqrt{\bar{\alpha}_t} \cdot \hat{\mathbf{x}}_0(\mathbf{x}_t)}{\sqrt{1 - \bar{\alpha}_t}}\right). \tag{7}$$

Unlike DDPM, the forward process defined by Equation (7) is deterministic; the value $p_\theta(\mathbf{x}_0)$ is entirely determined by $\mathbf{x}_T \sim \mathcal{N}(0, \mathbf{I})$ thus making DDIM an implicit model.

With a slight modification of the DDIM update, we are able to take larger denoising steps and accelerate inference. Given $\delta \geq 1$, we define an accelerated denoising process

$$\mathbf{x}_{t-\delta} = f_\theta^\delta(\mathbf{x}_t) = \sqrt{\bar{\alpha}_{t-\delta}} \cdot \hat{\mathbf{x}}_0(\mathbf{x}_t) + \sqrt{1 - \bar{\alpha}_{t-\delta}}\left(\frac{\mathbf{x}_t - \sqrt{\bar{\alpha}_t} \cdot \hat{\mathbf{x}}_0(\mathbf{x}_t)}{\sqrt{1 - \bar{\alpha}_t}}\right). \tag{8}$$

Using this process, inference is completed in just $T' \equiv T/\delta$ steps, albeit with degraded quality of the resulting sample $\mathbf{x}_0$ as $\delta$ becomes large.

## 4 Methods

We are interested in solving linear inverse problems of the form $\mathbf{y} = \mathbf{A}\mathbf{x}$, where $\mathbf{y} \in \mathbb{R}^d$ is a linear measurement of $\mathbf{x} \in \mathbb{R}^n$ and $\mathbf{A} \in \mathbb{R}^{d \times n}$ describes our measurement operator. For example, if $\mathbf{A} \in \{0, 1\}^{n \times n}$ is a binary mask (which is the case for, e.g., in-painting or sparse recovery problems) then $\mathbf{y}$ describes a partial measurement of $\mathbf{x}$. We seek an estimate $\hat{\mathbf{x}}$ that is consistent with our measurements: in the noiseless case, $\mathbf{A}\hat{\mathbf{x}} = \mathbf{y}$. More generally, we seek to recover a robust estimate of $\hat{\mathbf{x}}$ when the measurements $\mathbf{y}$ have been corrupted by Gaussian noise: $\mathbf{y} = \mathbf{A}\mathbf{x} + \boldsymbol{\sigma}$, where $\boldsymbol{\sigma} \sim \mathcal{N}(0, \sigma_y^2\mathbf{I})$.

In Section 4.1 we motivate the high-level approach of alternating unconditional DDIM steps with an adaptive number of gradient updates on the measurement residual energy $\nabla_{\mathbf{x}_t}\|\mathbf{A}\hat{\mathbf{x}}_0(\mathbf{x}_t) - \mathbf{y}\|^2$. We choose a step size and number of gradient descent steps to ensure that at each timestep $t$, the related quantity $\|\mathbf{A}\mathbf{x}_t - \mathbf{y}\|^2$ remains within a standard deviation of its expected value $\mathbb{E}_{\boldsymbol{\epsilon}_t}\left[\|\mathbf{A}\mathbf{x}_t - \mathbf{y}\|^2 \mid \mathbf{y}\right]$ under the forward process; as shown in Section 4.2, this ensures that the residual we are optimizing, $\|\mathbf{A}\hat{\mathbf{x}}_0(\mathbf{x}_t) - \mathbf{y}\|^2$, also remains in-distribution of the forward process with high probability. Finally, in Section 4.3 we discuss our algorithm for choosing a step size $\eta$ and stopping criteria based on a target residual energy. The full algorithm is described in Algorithm 1.

### 4.1 Optimizing $\hat{\mathbf{x}}_0(\mathbf{x}_t)$ to match the measurements

For linear measurements $\mathbf{A}$, the Tweedie formula for $\hat{\mathbf{x}}_0(\mathbf{x}_t)$ (and the corresponding plugin-estimate Equation (6)) extends to a formula for the expected measurements:

$$\mathbb{E}\left[\mathbf{y}|\mathbf{x}_t\right] = \mathbf{A}\mathbb{E}\left[\mathbf{x}_0|\mathbf{x}_t\right] \approx \mathbf{A}\hat{\mathbf{x}}_0(\mathbf{x}_t). \tag{9}$$

During the diffusion process, we would like the expected denoising trajectory to agree with the observation, i.e. $\mathbf{A}\hat{\mathbf{x}}_0(\mathbf{x}_t) = \mathbf{y}$. Therefore, a reasonable initial idea is to modify the DDIM updates Equation (7) to find the closest point $\mathbf{x}_{t-\delta}$ that satisfies the constraint $\mathbf{A}\hat{\mathbf{x}}_0(\mathbf{x}_{t-\delta}) = \mathbf{y}$. I.e., at each time step $t$, we force the Tweedie estimate of the posterior mean of $q(\mathbf{y}|\mathbf{x}_t)$ to match the observed measurements $\mathbf{y}$ while making the smallest possible update from our unconditional denoising step:

$$\begin{aligned} \underset{\mathbf{x}_{t-\delta}}{\arg\min} \quad & \|\mathbf{x}_{t-\delta} - f_\theta(\mathbf{x}_t)\|^2 \\ \text{subject to} \quad & \mathbf{A}\hat{\mathbf{x}}_0(\mathbf{x}_{t-\delta}) = \mathbf{y}. \end{aligned} \tag{10}$$

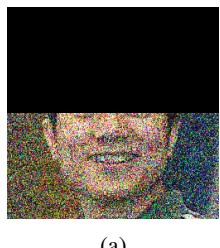 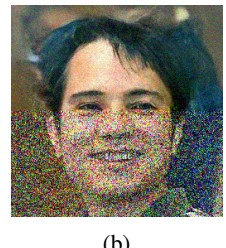 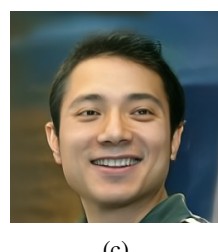

(a)  (b)  (c)

Figure 3: Results on a 50% noisy inpainting task with $\sigma_y = 0.2$. (a) is the noisy partial observation. (b) is generated by CDIM (Algorithm 1) without considering the Gaussian measurement noise, showing that we can **exactly** match the constraint even when the observation is out of distribution. (c) is generated by CDIM and using the values from Appendix B that consider the Gaussian measurement noise.

We face two conceptual challenges in optimizing Equation (10). First, for $t > 0$, typically no value $\mathbf{x}_t$ will satisfy $\mathbf{A}\hat{\mathbf{x}}_0(\mathbf{x}_t) = \mathbf{y}$ and therefore the optimization is infeasible. Second, the estimate of the score function $\nabla_{\mathbf{x}_t} \log q(\mathbf{x}_t)$ from the diffusion model, $\boldsymbol{\epsilon}_\theta(\mathbf{x}_t, t)$ may be inaccurate, particularly at large $t$; we risk overfitting to a bad plug-in estimate $\hat{\mathbf{x}}_0(\mathbf{x}_t)$ which will pull our iterate $\mathbf{x}_t$ far out of distribution during the denoising process. [3]

In light of these observations, we replace Equation (10) with a soft optimization

$$\arg\min_{\mathbf{x}_{t-\delta}} \|\mathbf{x}_{t-\delta} - f_\theta(\mathbf{x}_t)\|^2 + \lambda\|\mathbf{A}\hat{\mathbf{x}}_0(\mathbf{x}_{t-\delta}) - \mathbf{y}\|^2. \tag{11}$$

We can interpret Equation (11) as a relaxation of Equation (10) and we implement this optimization via gradient descent, initialized from $\mathbf{x}_{t-\delta}^{(0)} = f_\theta(\mathbf{x}_t)$ and computing $k$ gradient steps

$$\mathbf{x}_{t-\delta}^{(k)} = \mathbf{x}_{t-\delta}^{(k-1)} + \eta\nabla_{\mathbf{x}_{t-\delta}}\|\mathbf{A}\hat{\mathbf{x}}_0(\mathbf{x}_{t-\delta}) - \mathbf{y}\|^2. \tag{12}$$

The regularization by $\lambda\|\mathbf{A}\hat{\mathbf{x}}_0(\mathbf{x}_{t-\delta}) - \mathbf{y}\|^2$ is achieved implicitly by stopping after $k$ steps of gradient descent at a given timestep $t$. In contrast to a hard constraint at each timestep, this objective is robust to both (1) the possible infeasibility of $\mathbf{A}\hat{\mathbf{x}}_0(\mathbf{x}_{t-\delta}) = \mathbf{y}$ and (2) overfitting the measurements based on an inaccurate Tweedie plug-in estimator.

For notational purposes, we define $L_t$ as the energy of the measurement residual between the Tweedie posterior mean $\hat{\mathbf{x}}_0(\mathbf{x}_t)$ and our observation $\mathbf{y}$. This is the quantity we actively reduce via gradient descent with each projection step:

$$L_t := \|\mathbf{A}\hat{\mathbf{x}}_0(\mathbf{x}_t) - \mathbf{y}\|^2. \tag{13}$$

We can therefore interpret Equation (11) as a projection of the DDIM update $f_\theta(\mathbf{x}_t)$ onto the set of values $\mathbf{x}_{t-\delta}$ that sufficiently reduce the residual energy $L_{t-\delta}$. The full inference procedure is analogous to projected gradient descent, whereby we alternately take a step $f_\theta(\mathbf{x}_t)$ determined by the diffusion model, and then project back onto the set of plausible values for $L_{t-\delta}$ at timestep $t-\delta$. At high noise levels, the plausible domain is very large so our constraint is weak. As noise decreases, the plausible domain constricts to a radius of zero, achieving constraint satisfaction if we can ensure that $L_t = 0$ when $t = 0$.

As $t$ approaches 0, $\hat{\mathbf{x}}_0(\mathbf{x}_t)$ converges to $\mathbf{x}_t$ and $L_t = \|\mathbf{A}\hat{\mathbf{x}}_0(\mathbf{x}_t) - \mathbf{y}\|^2$ empirically behaves like a simple convex quadratic $\|\mathbf{A}\mathbf{x}_t - \mathbf{y}\|^2$, which can be minimized to arbitrary accuracy by taking sufficiently many gradient steps. This observation is why we can achieve exact recovery of the measurements $\mathbf{y} = \mathbf{A}\mathbf{x}_0$ in the final result $\mathbf{x}_0$. We demonstrate in Figure 3c that we can match a measurement even when it is out of distribution. We provide further calculations and empirical evidence to demonstrate the constraint convergence behavior in Appendix C.

---

[3]We illustrate both these claims by considering the Tweedie estimator Equation (6) in the case $t = T$. In this case, $\mathbf{x}_t \sim \mathcal{N}(0, I)$ is independent of $\mathbf{x}_0$ and therefore $\mathbb{E}[\mathbf{x}_0|\mathbf{x}_t] = \mathbb{E}[\mathbf{x}_0]$, the mean of the data distribution $q(\mathbf{x}_0)$. Unless $\mathbf{A}\mathbb{E}[\mathbf{x}_0] = \mathbf{y}$, the optimization is infeasible when $t = T$. Furthermore, we observe that when $t = T$, the plug-in estimator $\hat{\mathbf{x}}_0$ is not independent of $\mathbf{x}_t$ and $\hat{\mathbf{x}}_0 \neq \mathbb{E}[\mathbf{x}_0]$. This is indicative of error in the diffusion model, especially at high noise levels.

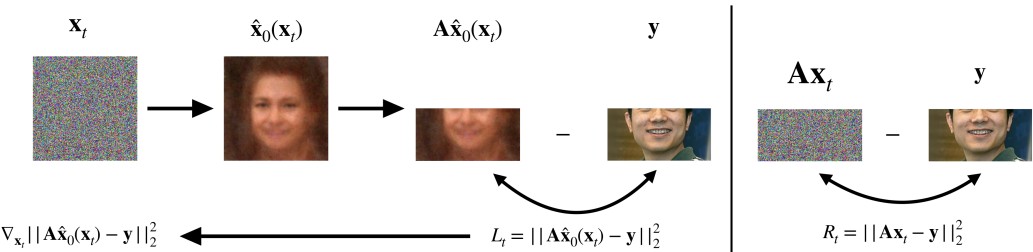

**Measurement Consistency Projection Step**  **Proxy Residual**

Figure 4: We show the conceptual overview of CDIM for a 50% inpainting task without measurement noise. (Left) We compute the Tweedie posterior estimate $\hat{\mathbf{x}}_0(\mathbf{x}_t)$ then apply the linear operator $\mathbf{A}$. This value $\mathbf{A}\hat{\mathbf{x}}_0(\mathbf{x}_t)$ is compared with the observation $\mathbf{y}$ to obtain our loss $L_t = \|\mathbf{A}\hat{\mathbf{x}}_0(\mathbf{x}_t) - \mathbf{y}\|^2$. We then update our iterate $\mathbf{x}_t$ with steps on $\nabla_{\mathbf{x}_t} L_t$. (Right) Our proxy residual $R_t = \|\mathbf{A}\mathbf{x}_t - \mathbf{y}\|^2$ has an anlytical $\chi^2$ distribution under the forward noising process and is used to guide the step size and number of steps for the left side process.

## 4.2 Maintaining $\|\mathbf{A}\mathbf{x}_t - \mathbf{y}\|^2$ Within its Plausible Region

When alternating unconditional diffusion steps with measurement consistency steps, a key design choice is how aggressively to minimize $L_t$. Specifically, at each timestep $t$, we must determine the domain of plausible values for the residual energy, and find the corresponding step size $\eta$ and number of gradient steps $k$ to minimize Equation (11). Keeping $L_t$ within its plausible region means we must sufficiently minimize $L_t$ at each timestep while also not pulling our current iterate $\mathbf{x}_t$ out of distribution of the unconditional diffusion process. Existing methods, such as DPS [2], typically apply a single small projection step per diffusion timestep, which insufficiently minimizes the residual energy when using accelerated DDIM schedules, leading to under-convergence (see Figure 7). Alternatively, fixing $k > 1$ across all timesteps is both computationally inefficient and may cause us to leave the plausible region, particularly at high noise levels where the residual is inherently large and uninformative.

To guide the optimization process more effectively, we propose to align the measurement residual with its expected behavior under the forward diffusion process. That is, at each reverse step $t \rightarrow t - \delta$, we would like to ensure that the observed residual $L_{t-\delta}$ is plausible under the distribution of residuals given by the forward process. This distribution can be defined formally via the forward noising process: Let $\varepsilon_t \sim \mathcal{N}(0, I)$ be the forward noise and consider $L_t = \|\mathbf{A}\,\hat{\mathbf{x}}_0(\mathbf{x}_t) - \mathbf{y}\|^2$ with $\mathbf{x}_t = \sqrt{\bar{\alpha}_t}\,\mathbf{x}_0 + \sqrt{1 - \bar{\alpha}_t}\,\varepsilon_t$ and $\mathbf{y} = \mathbf{A}\mathbf{x}_0$ for some true data sample $\mathbf{x}_0$. We then see that, under the forward diffusion process, $p(L_t|\mathbf{y})$ is a well defined distribution that depends on the forward noise and ground truth data distribution $\mathbf{x}_0$.

Our key idea is to compare the observed residual $L_t$ during the denoising process to its forward-process expectation $\mathbb{E}_{\varepsilon_t, \mathbf{x}_0}[L_t \mid \mathbf{y}]$ and adaptively adjust the optimization effort to maintain consistency. Intuitively, we want to keep our optimization target $L_t$ within a plausible region close to its expected value, which prevents under or over optimization of the projection steps. This idea is motivated by prior work [32, 3] that demonstrates improved generative performance when the reverse process mirrors the forward dynamics. Unfortunately, the forward expectation of $\|\mathbf{A}\hat{\mathbf{x}}_0(\mathbf{x}_t) - \mathbf{y}\|^2$ is intractable to compute as it depends on the true data distribution. Furthermore, every function evaluation $\hat{\mathbf{x}}_0$ (and therefore $L_t$) requires a neural network evaluation, which would make it computationally expensive to perform any kind of step size search even if we knew the plausible region..

Instead, we propose to use the distribution of a proxy residual $R_t := \|\mathbf{A}\mathbf{x}_t - \mathbf{y}\|^2$ to guide our projection steps. This residual has several desirable properties compared to $L_t$: it is analytically tractable under the forward diffusion process, inexpensive to compute, and strongly correlated with our target residual $L_t$. Specifically, we calculate in the Appendix A that under mild conditions, maintaining $R_t$ near its forward-process expectation, which we denote $\mu_t(\mathbf{y})$, ensures that $L_t$ is similarly controlled with high probability. This provides a computationally efficient and principled mechanism for adaptively controlling projection effort at each timestep, without requiring additional model evaluations. This entire procedure is shown at a high level in Figure 4.

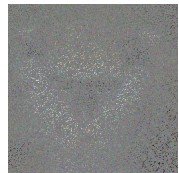 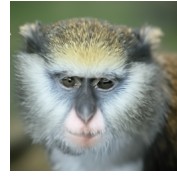 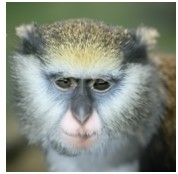 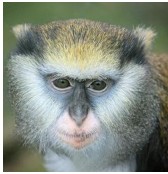

(a) Random in-painting task input $\mathbf{y}$  (b) Stopping at $\mu_R(\mathbf{y}) + 4\sigma_R$  (c) Stopping at $\mu_R + \sigma_R$  (d) Stopping at $\mu_R + 0.1\sigma_R$  (e) Ground Truth

Figure 5: We show the results of stopping at various points within the plausible region defined by $\mu_R(\mathbf{y}) + c \cdot \sigma_R(\mathbf{y})$ for different values of $c$. For $c < 1$ the results do not change dramatically, but numerically still improve (see Appendix D).

**Proposition 1** *Define the target residual energy $L_t := \|\mathbf{A}\hat{\mathbf{x}}_0(\mathbf{x}_t) - \mathbf{y}\|^2$ and the proxy residual energy $R_t := \|\mathbf{A}\mathbf{x}_t - \mathbf{y}\|^2$. Suppose that the proxy residual satisfies $|R_t - \mathbb{E}[R_t|\mathbf{y}]| \leq \gamma$. Then with high probability, $\left|L_t - \mathbb{E}[L_t \mid \mathbf{y}]\right| \leq \frac{\gamma}{\bar{\alpha}_t} + \mathcal{O}\left(\frac{\sqrt{1-\bar{\alpha}_t}}{\bar{\alpha}_t}\right)$.*

### 4.2.1 The $\chi^2$ Distribution of $R_t$

The distribution of the residual energy $p(\|\mathbf{A}\mathbf{x}_t - \mathbf{y}\|^2 \mid \mathbf{y})$ under the forward noising process follows a non-central generalized chi-squared distribution with tractable mean and variance:

$$\mu_t(\mathbf{y}) := \mathbb{E}_{\boldsymbol{\epsilon}_t}[R_t \mid \mathbf{y}] = (\sqrt{\bar{\alpha}_t} - 1)^2\|\mathbf{y}\|^2 + (1 - \bar{\alpha}_t)\operatorname{tr}(\mathbf{A}\mathbf{A}^\top), \tag{14}$$
$$\sigma_t^2(\mathbf{y}) := \operatorname{Var}_{\boldsymbol{\epsilon}_t}[R_t|y] = 2\operatorname{tr}((\boldsymbol{\Sigma}_t)^2) + 4\boldsymbol{\mu}_t^\top\boldsymbol{\Sigma}_t\boldsymbol{\mu}_t. \tag{15}$$

We derive this formally in Appendix B and also derive the corresponding mean and variance given noisy measurements $\mathbf{y} = \mathbf{A}\mathbf{x} + \mathcal{N}(0, \sigma_y^2\boldsymbol{I})$.

### 4.3 Choosing $\eta$ and Stopping Criteria

Now that we have an analytical understanding of our proxy residual energy $R_t := \|\mathbf{A}\mathbf{x}_t - \mathbf{y}\|^2$, we can determine the optimal projection step size $\eta$ and number of steps $k$ for each timestep $t$.

To decide when to stop projection steps, we compare the current residual energy $R_t$ to the forward expectation $\mu_t(\mathbf{y}) = \mathbb{E}_{\boldsymbol{\epsilon}_t}[R_t \mid \mathbf{y}]$ derived above. We denote the plausible region boundary as $\rho_t(\mathbf{y}) := \mu_t(\mathbf{y}) + c \cdot \sigma_t(\mathbf{y})$ where we aim to stay within some deviation of the mean. Lower values of $c$ typically produce better results at the tradeoff of more projection steps. We then halt the projection loop when the proxy residual lies within this plausible region boundary.

$$\|\mathbf{A}\mathbf{x}_{t-\delta} - \mathbf{y}\|^2 \leq \rho_{t-\delta}(\mathbf{y}). \tag{16}$$

This rule avoids both under- and over-projection while ensuring the denoising trajectory aligns with the expected residual behavior. We show comparative results of choosing a different stopping criteria $c$ in Figure 5 and in Section 5.3.

To determine the optimal step size $\eta$, we precompute the gradient of our objective:

$$\boldsymbol{g} := \nabla_{\mathbf{x}_{t-\delta}}\|\mathbf{A}\hat{\mathbf{x}}_0(\mathbf{x}_t) - \mathbf{y}\|^2. \tag{17}$$

We then perform a one-dimensional search (e.g., binary or line search) to select the $\eta$ that yields a future residual energy $\|\mathbf{A}(\mathbf{x}_{t-\delta} - \eta\boldsymbol{g}) - \mathbf{y}\|^2$ closest to the plausible region boundary $\rho_{t-\delta}$. We find empirically that bringing the residual back to the boundary of the plausible region is better than trying to bring it directly to the center of the plausible region.

$$\eta^* := \arg\min_{\eta\in\mathbb{R}}\left|\rho_{t-\delta}(\mathbf{y}) - \|\mathbf{A}(\mathbf{x}_{t-\delta} - \eta\boldsymbol{g}) - \mathbf{y}\|^2\right|. \tag{18}$$

Since the objective is a well-behaved quadratic function in $\eta$, this search is efficient and stable in practice. We summarize the complete algorithm in Algorithm 1, using a generalized step size $\delta$ during the denoising process.

**Algorithm 1** Constrained Diffusion Implicit Models

$\mathbf{x}_T \sim \mathcal{N}(\mathbf{0}, \mathbf{I})$
**for** $t = T, T - \delta.., 1$ **do**

$\quad \mathbf{x}_{t-\delta} \leftarrow \sqrt{\bar{\alpha}_{t-\delta}} \left( \frac{\mathbf{x}_t - \sqrt{1-\bar{\alpha}_t}\epsilon_\theta(\mathbf{x}_t, t)}{\sqrt{\bar{\alpha}_t}} \right) + \sqrt{1 - \bar{\alpha}_{t-\delta}}\epsilon_\theta(\mathbf{x}_t, t)$  $\quad \{\triangleright \text{ Unconditional Generation}\}$

$\quad \rho_t(\mathbf{y}) \leftarrow \mu_{t-\delta}(\mathbf{y}) + c \cdot \sigma_{t-\delta}(\mathbf{y})$  $\quad\quad\quad\quad\quad\quad\quad\quad\quad\quad\quad \{\triangleright \text{ Boundary of Plausible Region}\}$

$\quad$ **while** $\|\mathbf{A}\mathbf{x}_{t-\delta} - \mathbf{y}\|^2 > \rho_t(\mathbf{y})$ **do**

$\quad\quad \hat{\mathbf{x}}_0 \leftarrow \frac{1}{\sqrt{\bar{\alpha}_{t-\delta}}} \left( \mathbf{x}_{t-\delta} - \sqrt{1 - \bar{\alpha}_{t-\delta}}\epsilon_\theta(\mathbf{x}_{t-\delta}, t - \delta) \right)$

$\quad\quad \boldsymbol{g} \leftarrow \nabla_{\mathbf{x}_{t-\delta}} \|\mathbf{A}\hat{\mathbf{x}}_0(\mathbf{x}_{t-\delta}) - \mathbf{y}\|_2^2$

$\quad\quad \eta^* := \arg\min_{\eta \in \mathbb{R}} \left| \rho_t(\mathbf{y}) - \|\mathbf{A}(\mathbf{x}_{t-\delta} - \eta\boldsymbol{g}) - \mathbf{y}\|^2 \right|$

$\quad\quad \mathbf{x}_{t-\delta} \leftarrow \mathbf{x}_{t-\delta} - \eta^*\boldsymbol{g}$

$\quad$ **end while**
**end for**

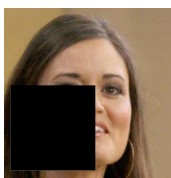 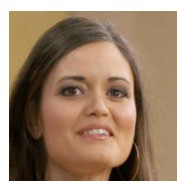 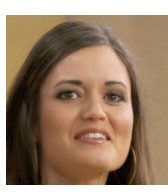 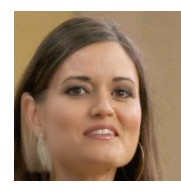 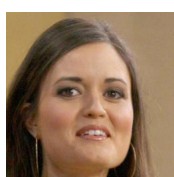

(a) Box inpainting task input $\mathbf{y}$ | (b) T'=25 (31 Total Projection Steps) | (c) T'=10 (15 Total Projection Steps) | (d) T'=4 (10 Total Projection Steps) | (e) Ground Truth

Figure 6: We show how the results change as we decrease the number of denoising steps $T'$. Notably, we still produce reasonable results with only $T' = 4$ denoising steps and a corresponding 10 total projection steps, for a total of 14 Neural Function Evaluations (NFEs) and $< 1$ second inference.

### 4.4 Poisson Noise

Possion noise is non-additive noise defined by $s\mathbf{y} \sim \text{Poisson}(s\mathbf{A}\mathbf{x})$, where $\mathbf{y}$ is interpreted as discrete integer pixel values. The scaling factor $s \leq 1$ controls the degree of Poisson noise. Poisson noise is not identically distributed across $\mathbf{y}$; the variance increases with the scale of each measurement. To remedy this, we consider the Pearson residuals [33]:

$$R(\mathbf{A}\hat{\mathbf{x}}_0, \mathbf{y}) = \frac{\lambda(\mathbf{y} - \mathbf{A}\hat{\mathbf{x}}_0)}{\sqrt{\lambda\hat{\mathbf{x}}_0}}. \tag{19}$$

These residuals are identically distributed; moreover, they are approximately normal $r \sim \mathcal{N}(0, 1)$ [34]. We can therefore treat the Pearson residuals as standard normal noise and solve the inverse problems using the same method for Gaussian measurement noise. Although the Pearson residuals closely follow the standard normal distribution for positive values of $\hat{\mathbf{x}}_0$, this breaks down for values of $\hat{\mathbf{x}}_0$ close to zero, and extreme noise levels $s$. In practice we find the Gaussian assumption to be valid for natural images corrupted by as much noise as $s \approx 0.025$. In Figure 1 we show an example of denoising an image corrupted by Poisson noise with $s = 0.05$.

## 5 Results and Experiments

We conduct experiments to demonstrate the efficiency and quality of CDIM across various tasks and datasets. In Section 5.1, we present quantitative comparisons to state-of-the-art approaches, followed by a comparison against DPS using DDIM in Section 5.2. In Section 5.3 we describe ablation studies examining inference speed and hyperparameters. Finally, in Section 5.4 we explore two novel applications of diffusion models for inverse problems.

Table 1: Quantitative results (FID, LPIPS) of our model and existing models on various linear inverse problems on FFHQ 256 × 256-1k validation dataset. (Lower is better). The best result is in **bold** and the second best is underlined.

| **FFHQ** | **Super Res** | | **Inpainting (box)** | | **Gaussian Deblur** | | **Inpainting (random)** | | **Runtime (seconds)** |
|---|---|---|---|---|---|---|---|---|---|
| Methods | FID | LPIPS | FID | LPIPS | FID | LPIPS | FID | LPIPS | |
| Ours $T' = 25$ | 33.87 | 0.276 | 27.51 | 0.1872 | 34.18 | 0.276 | 29.67 | 0.243 | 2.4 |
| Ours $T' = 50$ | 31.54 | 0.269 | **26.09** | 0.196 | **29.68** | **0.252** | 28.52 | 0.240 | 6.4 |
| FPS-SMC | **26.62** | **0.210** | 26.51 | **0.150** | 29.97 | 0.253 | 33.10 | 0.275 | 116.90 |
| DPS | 39.35 | 0.214 | 33.12 | 0.168 | 44.05 | 0.257 | **21.19** | **0.212** | 70.42 |
| DDRM | 62.15 | 0.294 | 42.93 | 0.204 | 74.92 | 0.332 | 69.71 | 0.587 | 2.0 |
| MCG | 87.64 | 0.520 | 40.11 | 0.309 | 101.2 | 0.340 | 29.26 | 0.286 | 73.2 |
| PnP-ADMM | 66.52 | 0.353 | 151.9 | 0.406 | 90.42 | 0.441 | 123.6 | 0.692 | 3.595 |
| Score-SDE | 96.72 | 0.563 | 60.06 | 0.331 | 109.0 | 0.403 | 76.54 | 0.612 | 32.39 |
| ADMM-TV | 110.6 | 0.428 | 68.94 | 0.322 | 186.7 | 0.507 | 181.5 | 0.463 | - |

## 5.1 Numerical Results on FFHQ and ImageNet

We evaluate CDIM on the FFHQ-1k [35] and ImageNet-1k [36] validation sets. Each dataset contains 256 × 256 RGB images scaled to the range [0, 1]. The tasks include 4x super-resolution, box inpainting, Gaussian deblur, and random inpainting. Details of each task are included in the appendix. For all tasks, we apply zero-centered Gaussian measurement noise with $\sigma = 0.05$. To ensure fair comparisons, use identical pre-trained diffusion models used in the baseline methods: for FFHQ we use the network from [2] and for ImageNet we use the network from [19]. We report Frechet Inception Distance (FID) [37] and Learned Perceptual Image Patch Similarity (LPIPS) [38] with peak signal-to-noise ratio (PSNR) results in Appendix E.5. All experiments are carried out on a single Nvidia A100 GPU.

In Table 1 we compare CDIM with several other inverse solvers using the FID and LPIPS metrics on the FFHQ dataset. We present results using our method with both $T' = 25$ denoising steps and $T' = 50$ denoising steps. In all cases we use $c = 0.1$ for the number of standard deviations in the stopping criteria. For ImageNet results please see Appendix E.4.

## 5.2 Additional Comparisons

We show a qualitative comparison against DPS [2] when we combine it DDIM and fewer steps (see Figure 7). We use the core DPS sampling algorithm, but with DDIM as the denoising algorithm instead of DDPM. The number of denoising steps is set to 15 (CDIM is limited to 15 projection steps) and the step size of DPS is increased in panel (c) to achieve the best constraint convergence

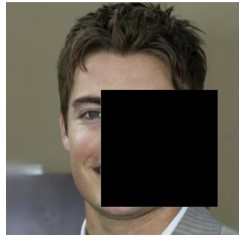 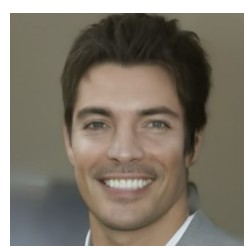 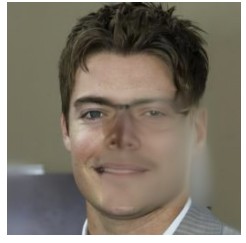 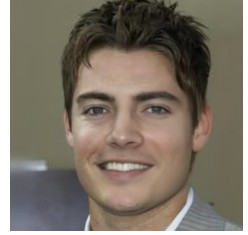

(a) Box inpainting without noise: input $\mathbf{y}$.    (b) DPS + DDIM MAE: 4%    (c) DPS + DDIM (large $\eta$) MAE: 0.3%    (d) CDIM MAE: 0.05%*

Figure 7: We show comparisons against DPS using DDIM sampling where all algorithms use 30 NFEs. For each output, we report the mean absolute error of the **observed** pixels, $\|\mathbf{Ax} - \mathbf{y}\|$, relative to the input. (b) If you simply run DPS with DDIM sampling, the constraint is not met. You can see this in the hair, which is blurrier, along with the background which has changed. (c) If you try to increase the step size of DPS, the observed region matches better, but the results diverge. (c) CDIM achieves strong constraint satisfcaation and inpainting results. *The MAE of CDIM should be 0, but the diffusion schedule sets $\overline{\alpha}_0 = 0.999$ for numeric stability.

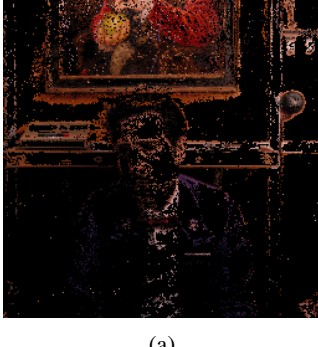 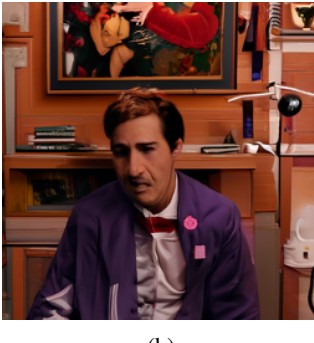

(a)                                                (b)

Figure 8: Noisy inpainting for sparse point cloud reprojection. (a) Shows a sparse point cloud projected to a desired camera angle. (b) Shows the result after our method is used for noisy inpainting.

possible. Note that when using DPS, no learning rate can lead to measurement consistency with the accelerated DDIM sampling scheduling without diverging.

We also compare against DAPs-50 [18], DSG [28], and Diff-PIR [25] in Appendix F.

### 5.3 Ablation Studies

CDIM only contains two hyperparameters: the number of denoising steps $T'$ and the plausible region stopping criteria constant $c$ multiplied by $\sigma_t(\mathbf{y})$. In Figure 6 we show how the results and total required projection steps change as we decrease the $T'$ to as few as 4 denoising steps. For the stopping criteria, Figure 5 shows qualitative results when changing $c$, and in Appendix D we show how quantitative results and number of projection steps change with $c$.

### 5.4 Additional Applications

**Time-Travel Rephotography** In Figure 1 we showcase an application of time-travel rephotography [39]. Antique cameras lack red light sensitivity, exaggerating wrinkles by filtering out skin subsurface scatter which occurs mostly in the red channel. To address this, we input the observed image into the blue color channel and use the pretrained FFHQ model with Algorithm 1 to project the face into the space of modern images. We further emphasize the power of our approach; [39] trained a specialized model for this task while we are able to use a pretrained model without modification.

**Sparse Point Cloud Reprojection** Twenty different images from a scene in The Grand Budapest Hotel scene were entered into Colmap [40] to generate a sparse 3D point cloud. Projections of this point cloud have roughly $90\%$ of the pixels missing. Furthermore, the measurements often contain significant amounts of non-Gaussian noise due to false correspondences. We can formulate this as noisy inpainting problem and use Algorithm 1 along with a variance threshold that adequately captures the imprecise nature of the point cloud. We showcase the results in Figure 8.

## 6 Conclusion

In this paper we introduced CDIM, a new approach for accelerating noisy linear inverse recovery using pretrained diffusion models. By projecting DDIM steps into a plausible region of of the forward process, we can enforce constraints without making out-of-distribution edits to the noised iterates $\mathbf{x}_t$. Note that our method cannot handle non-linear constraints because for a non-linear function $h$, $\mathbb{E}[h(\mathbf{x}_0)] \neq h(\mathbb{E}[\mathbf{x}_0])$. Therefore, we cannot extend Tweedie's estimate of the posterior mean $\mathbf{x}_0$ to an estimate of the posterior mean of non-linear observations $h(\mathbf{x}_0)$. However, for linear constraints, our method generates high quality images with faster inference than previous methods, creating a new point on the Pareto-frontier of quality vs. efficiency for linear inverse problems.

## Acknowledgments

This work was supported by the UW Reality Lab, Lenovo, Meta, Google, OPPO, and Amazon. We also thank Tatsunori Hashimoto for fruitful discussions in the early development of this work.

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

# A Proposition: Controlling the Proxy Residual Controls the True Objective

**Setup and notation.** Fix a reverse–diffusion step $t \in \{1, \ldots, T\}$ with cumulative noise level $\bar{\alpha}_t \in (0, 1)$. Let

$$\text{One–step variance:} \quad b := 1 - \bar{\alpha}_t$$

$$a := \sqrt{\bar{\alpha}_t} - 1,$$

$$\Sigma := \mathbf{A}\mathbf{A}^\top \in \mathbb{R}^{d \times d}.$$

Given the current state $\mathbf{x}_t = \sqrt{\bar{\alpha}_t}\,\mathbf{x}_0 + \sqrt{b}\,\boldsymbol{\epsilon}_t$ with $\boldsymbol{\epsilon}_t \sim \mathcal{N}(\mathbf{0}, \mathbf{I})$, define

$$\text{Proxy residual:} \quad \boldsymbol{r}_t := \mathbf{A}\mathbf{x}_t - \mathbf{y}, \quad R_t := \|\boldsymbol{r}_t\|^2,$$

$$\text{Mean of proxy residual:} \quad \mu_t(\mathbf{y}) := \mathbb{E}_{\boldsymbol{\epsilon}_t}[R_t \mid \mathbf{y}]$$

$$\text{Tweedie estimate:} \quad \hat{\mathbf{x}}_0 := \frac{\mathbf{x}_t - \sqrt{b}\,\boldsymbol{\epsilon}_\theta(\mathbf{x}_t, t)}{\sqrt{\bar{\alpha}_t}}.$$

$$\text{true objective:} \quad L_t := \|\mathbf{A}\hat{\mathbf{x}}_0 - \mathbf{y}\|^2,$$

$$\text{model noise in measurement space:} \quad \boldsymbol{\zeta}_t := \mathbf{A}\boldsymbol{\epsilon}_\theta(\mathbf{x}_t, t).$$

Throughout we assume the denoiser is *conditionally unbiased*: $\mathbb{E}\left[\boldsymbol{\epsilon}_\theta(\mathbf{x}_t, t) \mid \mathbf{x}_t\right] = \boldsymbol{\epsilon}_t$.

**Proposition Statement:** Suppose that the proxy residual is close to its expected value: $|R_t - \mu_t(\mathbf{y})| \leq \gamma$. Then with high probability, $|L_t - \mathbb{E}[L_t \mid \mathbf{y}]| \leq \frac{\gamma}{\bar{\alpha}_t} + \mathcal{O}(\frac{\sqrt{(1-\bar{\alpha}_t)}}{\bar{\alpha}_t})$, i.e. the target residual is close to its expectation.

**Step 1: Write $L_t$ in terms of $R_t$**

Multiply $\hat{\mathbf{x}}_0$ by $\mathbf{A}$ and subtract $\mathbf{y}$:

$$\mathbf{A}\hat{\mathbf{x}}_0 - \mathbf{y} = \frac{\mathbf{A}\mathbf{x}_t - \sqrt{b}\,\boldsymbol{\zeta}_t - \sqrt{\bar{\alpha}_t}\,\mathbf{y}}{\sqrt{\bar{\alpha}_t}} = \frac{\boldsymbol{r}_t - \sqrt{b}\,\boldsymbol{\zeta}_t - a\,\mathbf{y}}{\sqrt{\bar{\alpha}_t}}.$$

Squaring the norm yields:

$$L_t = \frac{1}{\bar{\alpha}_t}\left(R_t + b\|\boldsymbol{\zeta}_t\|^2 + a^2\|\mathbf{y}\|^2 - 2\sqrt{b}\langle\boldsymbol{r}_t, \boldsymbol{\zeta}_t\rangle - 2a\langle\boldsymbol{r}_t, \mathbf{y}\rangle + 2a\sqrt{b}\langle\boldsymbol{\zeta}_t, \mathbf{y}\rangle\right). \quad (20)$$

**Step 2: Conditional expectation.** Taking $\mathbb{E}[\cdot \mid \mathbf{y}]$ in Equation (20), using $\mathbb{E}[\langle\boldsymbol{r}_t, \boldsymbol{\zeta}_t\rangle \mid \mathbf{y}] = \mathbb{E}[\langle\boldsymbol{\zeta}_t, \mathbf{y}\rangle \mid \mathbf{y}] = 0$ (given the unbiased-score assumption and independence of $\boldsymbol{\epsilon}_t$ from $\mathbf{y}$), and using $\mathbb{E}[\boldsymbol{r}_t \mid \mathbf{y}] = \mathbb{E}[\sqrt{\bar{\alpha}_t}\mathbf{A}\mathbf{x}_0 + \sqrt{b}\mathbf{A}\varepsilon_t - \mathbf{y}] = a\,\mathbf{y}$ gives:

$$\mathbb{E}[L_t \mid \mathbf{y}] = \frac{1}{\bar{\alpha}_t}\left(\mu_t(\mathbf{y}) + b\,\mathrm{tr}\,\Sigma + a^2\|y\|^2 - 2a^2\|y\|^2\right)$$

$$= \frac{1}{\bar{\alpha}_t}\left(\mu_t(\mathbf{y}) + b\,\mathrm{tr}\,\Sigma - a^2\|y\|^2\right).$$

**Step 3: Decompose the deviation.** Now we can calculate how far our true objective deviates from its expected value. Write

$$\Delta_1 := \|\boldsymbol{\zeta}_t\|^2 - \mathrm{tr}\,\Sigma, \quad \Delta_2 := \langle\boldsymbol{r}_t, \boldsymbol{\zeta}_t\rangle, \quad \Delta_3 := \langle\boldsymbol{\zeta}_t, y\rangle, \quad \Delta_4 := \langle\boldsymbol{r}_t, y\rangle - a\|y\|^2.$$

Then

$$\left|L_t - \mathbb{E}[L_t \mid y]\right| = \frac{1}{\bar{\alpha}_t}\left(|R_t - \mu_t(y)| + b|\Delta_1| + 2\sqrt{b}\,|\Delta_2| + 2|a|\sqrt{b}\,|\Delta_3| + 2|a|\,|\Delta_4|\right). \quad (21)$$

The first term in Equation (21) is the deviation of the proxy residual from its mean, which satisfies

$$|\,R_t - \mu_t(\mathbf{y})\,| \;\leq\; \gamma.$$

We then use Hanson-Wright and Cauchy-Schwarz to bound the remainder of the terms.

**Step 4: Concentration bounds for the $\Delta_i$.**

$\Delta_1$: Since $\boldsymbol{\zeta}_t = A\boldsymbol{\epsilon}_t$ with $\boldsymbol{\epsilon}_t \sim \mathcal{N}(\mathbf{0}, \mathbf{I})$, $\|\boldsymbol{\zeta}_t\|^2$ is a quadratic form in a Gaussian vector. Hanson–Wright [41] gives

$$|\Delta_1| \;\leq\; C_1\sqrt{\operatorname{tr}(\Sigma^2)}$$

with probability at least $1 - 2e^{-c_1 d}$. For measurement space dimension $d$: $\mathbf{y} \in \mathbb{R}^d$ and constant $c_1 > 0$.

$\Delta_2$: Using Cauchy-Schwarz, a $\chi^2$ tail for $\|\boldsymbol{\zeta}_t\|$ and the given bound $\|\boldsymbol{r}_t\| \leq \sqrt{\mu_t(y) + \gamma}$,

$$|\Delta_2| \leq \|\boldsymbol{r}_t\|\,\|\boldsymbol{\zeta}_t\| \leq \sqrt{\mu_t(y) + \gamma}\; C_2\sqrt{\operatorname{tr}\Sigma}.$$

$\Delta_3$: We can use a standard Gaussian tail bound since $\langle \boldsymbol{\zeta}_t, y\rangle \sim \mathcal{N}\big(0,\, y^\top \Sigma y\big)$. This yields

$$|\Delta_3| \leq C_2\,\|y\|\,\sqrt{\operatorname{tr}\Sigma}.$$

$\Delta_4$: This is a deterministic bound once $\|\boldsymbol{r}_t\|$ is bounded. Using $|a| = \sqrt{\bar{\alpha}_t} - 1 \leq \sqrt{b}$ and $\|\boldsymbol{r}_t\| \leq \sqrt{\mu_t(y) + \gamma}$,

$$|\Delta_4| = \left|\langle r_t, y\rangle - a\|y\|^2\right| \leq \|\boldsymbol{r}_t\|\,\|y\| + |a|\,\|y\|^2 \leq \sqrt{\mu_t(y) + \gamma}\,\|y\| + \sqrt{b}\,\|y\|^2.$$

Each of the three genuinely probabilistic bounds occurs with failure probability $2e^{-cd}$ for measurement space dimension $d$: $\mathbf{y} \in \mathbb{R}^d$ and absorbing constants into $c > 0$. In typical proofs involving Hanson-Wright, $c_i \approx 10^{-2}$ [42], and since $d$ is typically much larger than 100, the failure probability remains exceedingly small.

**Step 5: Assemble the pieces.**

Insert the bounds for $\Delta_{1:4}$ into Equation (21), use $\sqrt{\mu_t(y) + \gamma} \leq \sqrt{\mu_t(y)} + \sqrt{\gamma}$, and absorb numerical constants into a universal $C > 0$:

$$\left|L_t - \mathbb{E}[L_t \mid y]\right| \;\leq\; \frac{\gamma}{\bar{\alpha}_t} + \frac{C\sqrt{b}}{\bar{\alpha}_t}\left(\sqrt{\gamma} + \|y\|\right) + \frac{C\,b}{\bar{\alpha}_t}\,\|y\|^2.$$

**Step 6: Conclusion.**

$$|\,R_t - \mu_t(\mathbf{y})| \leq \gamma \;\implies\; \left|L_t - \mathbb{E}[L_t \mid y]\right| \;\leq\; \frac{\gamma}{\bar{\alpha}_t} \;+\; \mathcal{O}\!\left(\tfrac{\sqrt{1-\bar{\alpha}_t}}{\bar{\alpha}_t}\left[\,\sqrt{\gamma} + \|y\|\,\right] + \tfrac{1-\bar{\alpha}_t}{\bar{\alpha}_t}\,\|y\|^2\right),$$

with probability at least $1 - 6e^{-cd}$. Because $1 - \bar{\alpha}_t = b \ll 1$ for all practical timesteps, the additional terms are dominated by $\sqrt{b}/\bar{\alpha}_t$, leaving

$$\boxed{\left|L_t - \mathbb{E}[L_t \mid y]\right| \;\leq\; \frac{\gamma}{\bar{\alpha}_t} + \mathcal{O}\!\left(\tfrac{\sqrt{1-\bar{\alpha}_t}}{\bar{\alpha}_t}\right)},$$

as claimed. Notice that As $t \to 0$ ($\bar{\alpha}_t \to 1$) the second term vanishes, so matching the proxy residual to its mean immediately controls the true objective with the same statistical precision.

# B   Distribution of $\|\mathbf{A}\mathbf{x}_t - \mathbf{y}\|^2$ with Noiseless and Noisy Observations

## B.1   Noiseless Observations

Let us define our proxy residual vector $\boldsymbol{r}_t := \mathbf{A}\mathbf{x}_t - \mathbf{y}$ and the energy $R_t := \|\boldsymbol{r}_t\|^2 = \|\mathbf{A}\mathbf{x}_t - \mathbf{y}\|^2$.

We start with the forward diffusion equation in Equation (2) and multiply both sides by $\mathbf{A}$. We then subtract $\mathbf{y}$ from both sides, yielding the residual

$$\boldsymbol{r}_t := \mathbf{A}\mathbf{x}_t - \mathbf{y} = (\sqrt{\bar{\alpha}_t} - 1)\mathbf{y} + \sqrt{1 - \bar{\alpha}_t}\mathbf{A}\boldsymbol{\epsilon}_t, \quad \boldsymbol{\epsilon}_t \sim \mathcal{N}(\mathbf{0}, \mathbf{I}). \tag{22}$$

This implies that $p(\boldsymbol{r}_t|\mathbf{y}) \sim \mathcal{N}(\boldsymbol{\mu}_t(y), \boldsymbol{\Sigma}_t)$, where

$$\boldsymbol{\mu}_t(y) = (\sqrt{\bar{\alpha}_t} - 1)\mathbf{y},$$
$$\boldsymbol{\Sigma}_t = (1 - \bar{\alpha}_t)\mathbf{A}\mathbf{A}^\top.$$

The residual energy $R_t := \|\mathbf{A}\mathbf{x}_t - \mathbf{y}\|^2 = \|\boldsymbol{r}_t\|^2$ is thus a chi-squared distribution with the following mean $\mu_t(\mathbf{y})$ and variance $\sigma_t^2(\mathbf{y})$:

$$\mu_t(\mathbf{y}) := \mathbb{E}_{\boldsymbol{\epsilon}_t}[R_t \mid \mathbf{y}] = (\sqrt{\bar{\alpha}_t} - 1)^2\|\mathbf{y}\|^2 + (1 - \bar{\alpha}_t)\operatorname{tr}(\mathbf{A}\mathbf{A}^\top), \tag{23}$$
$$\sigma_t^2(\mathbf{y}) := \operatorname{Var}_{\boldsymbol{\epsilon}_t}[R_t|y] = 2\operatorname{tr}((\boldsymbol{\Sigma}_t)^2) + 4\boldsymbol{\mu}_t^\top\boldsymbol{\Sigma}_t\boldsymbol{\mu}_t. \tag{24}$$

Note that often our linear operator $\mathbf{A}$ is implemented in a functional form, where computing the actual matrix representation or its transpose is inconvenient. In practice, we can use trace estimators such as Hutchinson's method [43] to estimate all required values involving $\mathbf{A}$, such as $\operatorname{tr}(\mathbf{A}\mathbf{A}^\top)$, $\operatorname{tr}((\boldsymbol{\Sigma}_t^r)^2)$, and $\boldsymbol{\mu}_t^\top\boldsymbol{\Sigma}_t\boldsymbol{\mu}_t$ using only matrix-vector products.

## B.2   Noisy Observations

We derive the mean and variance of our residual energy $R_t := \|\mathbf{A}\mathbf{x}_t - \mathbf{y}\|^2$ during the forward diffusion process when we have measurement noise: $\mathbf{y} = \mathbf{A}\mathbf{x}_0 + \boldsymbol{\sigma_y}$ where $\boldsymbol{\sigma_y} \sim \mathcal{N}(0, \sigma_y^2\boldsymbol{I})$.

**Unbiased substitutions.**   Because we never observe the latent projection $\mathbf{A}\mathbf{x}_0$, we replace its quadratic forms with statistics that depend only on the noisy measurement $\mathbf{y}$ and $d := \dim(\mathbf{y})$:

$$\|\mathbf{A}\mathbf{x}_0\|^2 = \|\mathbf{y}\|^2 - d\sigma_y^2, \tag{25}$$
$$(\mathbf{A}\mathbf{x}_0)^\top\Sigma(\mathbf{A}\mathbf{x}_0) = \mathbf{y}^\top\Sigma\mathbf{y} - \sigma_y^2\operatorname{tr}\Sigma, \tag{26}$$

with $\Sigma := \mathbf{A}\mathbf{A}^\top$.

**First two moments of the residual energy.**   Using the forward-diffusion decomposition $\mathbf{x}_t = \sqrt{\bar{\alpha}_t}\,\mathbf{x}_0 + \sqrt{1 - \bar{\alpha}_t}\,\boldsymbol{\epsilon}_t$, $\boldsymbol{\epsilon}_t \sim \mathcal{N}(\mathbf{0}, \mathbf{I})$, one finds the residual $\mathbf{A}\mathbf{x}_t - \mathbf{y} = (\sqrt{\bar{\alpha}_t} - 1)\mathbf{A}\mathbf{x}_0 + \sqrt{1 - \bar{\alpha}_t}\,\mathbf{A}\boldsymbol{\epsilon}_t - \boldsymbol{\sigma}_y$. After substituting the unbiased identities Equation (25)–Equation (26) and taking expectations over both noise sources $\boldsymbol{\epsilon}_t$ and $\boldsymbol{\sigma}_y$, we obtain closed-form expressions that are fully observable.

**Expectation.**

$$\mathbb{E}[R_t \mid \mathbf{y}] = \underbrace{(1 - \bar{\alpha}_t)\operatorname{tr}(\Sigma)}_{\text{diffusion noise}} + \underbrace{m\sigma_y^2\big[1 - (\sqrt{\bar{\alpha}_t} - 1)^2\big]}_{\text{measurement noise}} + \underbrace{(\sqrt{\bar{\alpha}_t} - 1)^2\|\mathbf{y}\|^2}_{\text{deterministic bias}}. \tag{27}$$

**Variance.**   Writing $\tilde{\Sigma}_t = (1 - \bar{\alpha}_t)\Sigma + \sigma_y^2\mathbf{I}$ and $\tilde{\boldsymbol{\mu}}_t = (\sqrt{\bar{\alpha}_t} - 1)\mathbf{y}$, the non-central $\chi^2$ moment formula $\operatorname{Var}(Q_t) = 2\operatorname{tr}(\tilde{\Sigma}_t^2) + 4\tilde{\boldsymbol{\mu}}_t^\top\tilde{\Sigma}_t\tilde{\boldsymbol{\mu}}_t$ gives

$$\operatorname{Var}[R_t \mid \mathbf{y}] = 2\Big[(1 - \bar{\alpha}_t)^2\operatorname{tr}(\Sigma^2) + 2(1 - \bar{\alpha}_t)\sigma_y^2\operatorname{tr}\Sigma + m\sigma_y^4\Big]$$
$$+ 4(\sqrt{\bar{\alpha}_t} - 1)^2\Big[(1 - \bar{\alpha}_t)(\mathbf{y}^\top\Sigma\mathbf{y} - \sigma_y^2\operatorname{tr}\Sigma) + \sigma_y^2(\|\mathbf{y}\|^2 - m\sigma_y^2)\Big]. \tag{28}$$

## C Convergence of Constraint Satisfaction

We now analyze the convergence properties of the constraint satisfaction procedure in CDIM for the noiseless case. The algorithm alternates between unconditional diffusion updates and projection steps:

1. Unconditional update: $f_\theta(\mathbf{x}_t) = \text{DDIM.step}(\mathbf{x}_t)$
2. Projection:
   $\mathbf{x}_{t-\delta} = \arg\min_{\mathbf{x}_{t-\delta}} \|\mathbf{x}_{t-\delta} - f_\theta(\mathbf{x}_t)\|^2$ s.t. $\mathbf{A}\hat{\mathbf{x}}_0(\mathbf{x}_t) = \mathbf{y}$

where $\hat{\mathbf{x}}_0(\mathbf{x}_t)$ denotes the Tweedie estimate $\mathbb{E}[\mathbf{x}_0|\mathbf{x}_t]$ at timestep $t$. When the constraint is infeasible, we perform gradient descent on $\|\mathbf{A}\hat{\mathbf{x}}_0(\mathbf{x}_t) - \mathbf{y}\|^2$.

We first show that the Tweedie estimate converges to the identity mapping, and characterize its rate of convergence. We also demonstrate this rate of convergence empirically in Figure 9.

Given that, we show that as $t \to 0$, finding $\mathbf{x}_t$ such that $\|\mathbf{A}\hat{\mathbf{x}}_0(\mathbf{x}_t) - \mathbf{y}\|^2 = 0$ is feasible and satisfiable via the proposed gradient descent algorithm.

**Tweedie Convergence.** In this section, we show that as $t \to 0$, the Tweedie estimate converges to the identity mapping:

$$\sup_{\mathbf{x}_t} \|\hat{\mathbf{x}}_0(\mathbf{x}_t) - \mathbf{x}_t\|_2 \leq \varepsilon(t) \tag{29}$$

where $\varepsilon(t) \to 0$ as $t \to 0$.

Consider the forward diffusion process:

$$\mathbf{x}_t = \sqrt{1 - \bar{\beta}_t}\,\mathbf{x}_0 + \sqrt{\bar{\beta}_t}\,\boldsymbol{\epsilon}, \quad \boldsymbol{\epsilon} \sim \mathcal{N}(0, \mathbf{I}) \tag{30}$$

The Tweedie estimate (posterior mean) is given by:

$$\hat{\mathbf{x}}_0(\mathbf{x}_t) = \frac{\mathbf{x}_t}{\sqrt{1 - \bar{\beta}_t}} - \frac{\bar{\beta}_t}{\sqrt{1 - \bar{\beta}_t}}\nabla_{\mathbf{x}_t} \log p(\mathbf{x}_t) \tag{31}$$

For small $\bar{\beta}_t$ (as $t \to 0$), we perform a Taylor expansion:

$$\frac{1}{\sqrt{1 - \bar{\beta}_t}} = 1 + \frac{\bar{\beta}_t}{2} + O(\bar{\beta}_t^2) \tag{32}$$

$$\frac{\bar{\beta}_t}{\sqrt{1 - \bar{\beta}_t}} = \bar{\beta}_t\left(1 + \frac{\bar{\beta}_t}{2}\right) + O(\bar{\beta}_t^3) \tag{33}$$

Substituting into $\hat{\mathbf{x}}_0(\mathbf{x}_t)$:

$$\hat{\mathbf{x}}_0(\mathbf{x}_t) = \mathbf{x}_t\left(1 + \frac{\bar{\beta}_t}{2}\right) - \bar{\beta}_t\left(1 + \frac{\bar{\beta}_t}{2}\right)\nabla_{\mathbf{x}_t} \log p(\mathbf{x}_t) + O(\bar{\beta}_t^2) \tag{34}$$

$$= \mathbf{x}_t + \frac{\bar{\beta}_t}{2}\mathbf{x}_t - \bar{\beta}_t\nabla_{\mathbf{x}_t} \log p(\mathbf{x}_t) + O(\bar{\beta}_t^2) \tag{35}$$

The deviation from $\mathbf{x}_t$ is thus:

$$\hat{\mathbf{x}}_0(\mathbf{x}_t) - \mathbf{x}_t = \frac{\bar{\beta}_t}{2}\mathbf{x}_t - \bar{\beta}_t\nabla_{\mathbf{x}_t} \log p(\mathbf{x}_t) + O(\bar{\beta}_t^2) \tag{36}$$

Taking norms and applying the triangle inequality:

$$\|\hat{\mathbf{x}}_0(\mathbf{x}_t) - \mathbf{x}_t\|_2 \leq \frac{\bar{\beta}_t}{2}\|\mathbf{x}_t\|_2 + \bar{\beta}_t\|\nabla_{\mathbf{x}_t} \log p(\mathbf{x}_t)\|_2 + O(\bar{\beta}_t^2). \tag{37}$$

To estimate the scaling of $\|\nabla_{\mathbf{x}_t} \log p(\mathbf{x}_t)\|_2$, recall that the marginal $p(\mathbf{x}_t)$ is the Gaussian-smoothed density

$$p(\mathbf{x}_t) = (p_0 * \mathcal{N}(0, \bar{\beta}_t I))(\mathbf{x}_t),$$

whose log-gradient scales as the inverse of the noise standard deviation:

$$\|\nabla_{\mathbf{x}_t} \log p(\mathbf{x}_t)\|_2 = O\left(\frac{1}{\sqrt{\bar{\beta}_t}}\right)$$

with high probability. Intuitively, as $\bar{\beta}_t \to 0$, $p(\mathbf{x}_t)$ becomes sharply peaked around $\mathbf{x}_0$, and its log-density gradient increases proportionally to the inverse of the noise scale.

Substituting this scaling gives a dominant term of $\sqrt{\bar{\beta}_t}$:

$$\|\hat{\mathbf{x}}_0(\mathbf{x}_t) - \mathbf{x}_t\|_2 = O(\sqrt{\bar{\beta}_t}) \tag{38}$$

with high probability as $t \to 0$.

We show this empirically in Figure 9. This demonstrates that the Tweedie estimate of the posterior mean converges to the identity mapping at a rate proportional to $\sqrt{\bar{\beta}_t}$.

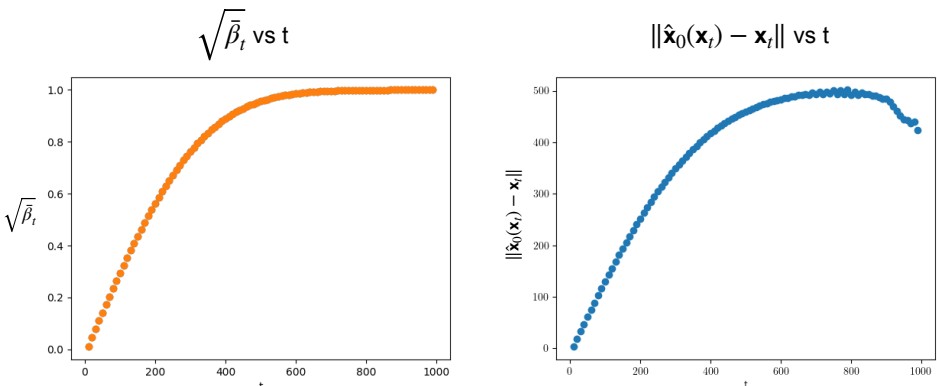

Figure 9: We show that $\|\hat{\mathbf{x}}_0(\mathbf{x}_t) - \mathbf{x}_t\|_2 = O(\sqrt{\bar{\beta}_t})$, demonstrating that the Tweedie estimate of the posterior mean converges to the identity mapping as $t \to 0$. (Left) plots $\sqrt{\bar{\beta}_t}$; (Right) plots $\|\hat{\mathbf{x}}_0(\mathbf{x}_t) - \mathbf{x}_t\|_2$ for a Gaussian deblur task. At higher $t$, model inaccuracies in estimating $\nabla_{\mathbf{x}_t} \log p(\mathbf{x}_t)$ cause deviations from the convergence pattern.

**Convergence of Constraint Satisfaction.** Based on the convergence of the Tweedie estimate to $\mathbf{x}_t$, we show that for sufficiently small $t$:

1. The constraint set $\{\mathbf{x}_t : \mathbf{A}\hat{\mathbf{x}}_0(\mathbf{x}_t) = \mathbf{y}\}$ is non-empty.
2. Gradient descent on $\|\mathbf{A}\hat{\mathbf{x}}_0(\mathbf{x}_t) - \mathbf{y}\|^2$ converges to a point satisfying the constraint.

First, we show that the optimization landscape becomes increasingly well-behaved as $t \to 0$. Consider the objective:

$$\|\mathbf{A}\hat{\mathbf{x}}_0(\mathbf{x}_t) - \mathbf{y}\|^2 = \|\mathbf{A}\mathbf{x}_t - \mathbf{y} + \mathbf{A}(\hat{\mathbf{x}}_0(\mathbf{x}_t) - \mathbf{x}_t)\|^2 \tag{39}$$
$$= \|\mathbf{A}\mathbf{x}_t - \mathbf{y}\|^2$$
$$+ 2\langle \mathbf{A}\mathbf{x}_t - \mathbf{y}, \mathbf{A}(\hat{\mathbf{x}}_0(\mathbf{x}_t) - \mathbf{x}_t)\rangle$$
$$+ \|\mathbf{A}(\hat{\mathbf{x}}_0(\mathbf{x}_t) - \mathbf{x}_t)\|^2. \tag{40}$$

By Tweedie Convergence, the second and third terms are bounded by $O(\varepsilon(t))$. Therefore, as $t \to 0$, the objective converges to the convex quadratic $\|\mathbf{A}\mathbf{x}_t - \mathbf{y}\|^2$, which can be efficiently optimized via gradient descent.

For feasibility, note that as $t \to 0$, finding $\mathbf{x}_t$ such that $\mathbf{A}\hat{\mathbf{x}}_0(\mathbf{x}_t) = \mathbf{y}$ becomes equivalent to finding $\mathbf{x}_t$ such that $\mathbf{A}\mathbf{x}_t = \mathbf{y}$ up to an error of $O(\varepsilon(t))$. The latter is feasible whenever $\mathbf{y}$ is in the range of $\mathbf{A}$, which is the standard assumption for linear inverse problems.

As $t \to 0$, $\varepsilon(t)$ approaches zero, making the constraint feasible. Moreover, since the objective approaches a convex quadratic, gradient descent will converge to the global minimum for sufficiently small $t$.

# D Ablation Studies

## D.1 Stopping Criteria

We consider variations of the stopping criteria where we use a different threshold besides 1 standard deviation on a **noiseless** random inpainting task when $T' = 25$. We consider three different stopping criteria and report the LPIPS on the FFHQ test set as well as the total projection steps during the inference. Notice that having a more generous stopping criteria does not always lead to fewer projection steps. We risk underfitting at early timesteps, and as the plausible region reduces to a radius of 0, we must take extra steps at later time steps to ensure consistency.

| Stopping criteria | Total Projection Steps | Random Inpainting LPIPS |
|---|---|---|
| $\mu_R + 3\sigma_R$ | 24 | 0.296 |
| $\mu_R + \sigma_R$ | 15 | 0.171 |
| $\mu_R + 0.5\sigma_R$ | 25 | 0.169 |

Table 2: Comparing stopping criteria based on standard deviations to the expected residual energy.

# E Additional Experimental Details

## E.1 Task Details

We describe additional details for each inverse task used in our experiments.

**Super Resolution** Images are downsampled to $64 \times 64$ using bicubic downsampling with a factor of $4$.

**Box Inpainting** A random box of size $128 \times 128$ is chosen uniformly within the image. Those pixels are masked out affected all three of the RGB channels.

**Gaussian Deblur** A Gaussian Kernel of size $61 \times 61$ and intensity 3 is applied to the entire image.

**Random Inpainting** Each pixel is masked out with probability $92\%$ affecting all three of the RGB channels

**50% Inpainting** In various figures, we showcase a $50\%$ inpainting task where the top half of an image is masked out. This task is more challenging than box inpainting and can better illustrate differences between results.

## E.2 Number of Projection Steps

Across a variety of tasks and dataset we find that $T' = 25$ denoising steps leads to an average of $27.2$ total projection steps and $T' = 50$ leads to an average of $46.5$ total projection steps. For experiments where we want to strictly limit the NFEs for fair comparison, we simply disallow additional projection steps after reaching the limit.

## E.3 Measuring Runtime

To measure wall-clock runtime, we used a single A100 and ran all the inverse problems (super-resolution, box inpainting, gaussian deblur, random inpainting) on the FFHQ dataset. We only consider the runtime of the algorithm, without considering the python initialization time, model loading, or image io. For each task, we measured the runtime on 10 images and averaged the result to produce the final result. We note that the baseline runtimes are taken from [15], where only the box inpainting task was considered. The runtime does not vary much between tasks when using CDIM, so we report our average runtime across tasks as a fair comparison metric.

## E.4 ImageNet Results

In Table 3 we report FID and LPIPS for ImageNet.

Table 3: Quantitative results (FID, LPIPS) of our model and existing models on various linear inverse problems on the Imagenet $256 \times 256$-1k validation dataset. (Lower is better)

| Imagenet | Super Resolution | | Inpainting (box) | | Gaussian Deblur | | Inpainting (random) | |
|---|---|---|---|---|---|---|---|---|
| Methods | FID | LPIPS | FID | LPIPS | FID | LPIPS | FID | LPIPS |
| Ours - T' = 25 | 53.70 | 0.378 | 52.00 | 0.267 | 56.10 | 0.393 | 51.96 | 0.370 |
| Ours - T' = 50 | 47.45 | 0.339 | 50.31 | 0.251 | 38.69 | 0.347 | 46.20 | 0.332 |
| FPS-SMC | 47.30 | 0.316 | 33.24 | 0.212 | 54.21 | 0.403 | 42.77 | 0.328 |
| DPS | 50.66 | 0.337 | 38.82 | 0.262 | 62.72 | 0.444 | 35.87 | 0.303 |
| DDRM | 59.57 | 0.339 | 45.95 | 0.245 | 63.02 | 0.427 | 114.9 | 0.665 |
| MCG | 144.5 | 0.637 | 39.74 | 0.330 | 95.04 | 0.550 | 39.19 | 0.414 |
| PnP-ADMM | 97.27 | 0.433 | 78.24 | 0.367 | 100.6 | 0.519 | 114.7 | 0.677 |
| Score-SDE | 170.7 | 0.701 | 54.07 | 0.354 | 120.3 | 0.667 | 127.1 | 0.659 |
| ADMM-TV | 130.9 | 0.523 | 87.69 | 0.319 | 155.7 | 0.588 | 189.3 | 0.510 |

## E.5 PSNR Results

See Tables 4 and 5

Table 4: Quantitative results (PSNR) of our model and existing models on various linear inverse problems on the FFHQ 256-1k validation dataset. (Higher is better)

| Imagenet | Super Resolution | Inpainting (box) | Gaussian Deblur | Inpainting (random) |
|---|---|---|---|---|
| Methods | PSNR | PSNR | PSNR | PSNR |
| Ours - T' = 25 | 27.08 | 23.20 | 26.77 | 26.49 |
| Ours - T' = 50 | 27.30 | 23.47 | 27.03 | 27.10 |
| FPS-SMC | 28.10 | 24.70 | 26.54 | 27.33 |
| DPS | 25.67 | 22.47 | 24.25 | 25.23 |
| DDRM | 25.36 | 22.24 | 23.36 | 9.19 |
| MCG | 20.05 | 19.97 | 6.72 | 21.57 |
| PnP-ADMM | 26.55 | 11.65 | 24.93 | 8.41 |
| Score-SDE | 17.62 | 18.51 | 7.21 | 13.52 |
| ADMM-TV | 23.86 | 17.81 | 22.37 | 22.03 |

Table 5: Quantitative results (PSNR) of our model and existing models on various linear inverse problems on the Imagenet $256 \times 256$-1k validation dataset. (Higher is better)

| Imagenet | Super Resolution | Inpainting (box) | Gaussian Deblur | Inpainting (random) |
|---|---|---|---|---|
| Methods | PSNR | PSNR | PSNR | PSNR |
| Ours - T' = 25 | 23.67 | 19.67 | 22.78 | 22.38 |
| Ours - T' = 50 | 23.92 | 20.06 | 23.32 | 22.61 |
| FPS-SMC | 24.78 | 22.03 | 23.81 | 24.12 |
| DPS | 23.87 | 18.90 | 21.97 | 22.20 |
| DDRM | 24.96 | 18.66 | 22.73 | 14.29 |
| MCG | 13.39 | 17.36 | 16.32 | 19.03 |
| PnP-ADMM | 23.75 | 12.70 | 21.81 | 8.39 |
| Score-SDE | 12.25 | 16.48 | 15.97 | 18.62 |
| ADMM-TV | 22.17 | 17.96 | 19.99 | 20.96 |

# F  Additional Comparisons

## F.1  Comparison with DSG

We show a qualitative comparison against DSG [28] on 3 tasks in Figure 10. We used the official code from their github, and generated results with 25 DDIM diffusion steps for both DSG and CDIM (and $K = 1$ for CDIM). As you can see, the DSG results are blurrier and sometimes contain artifacts

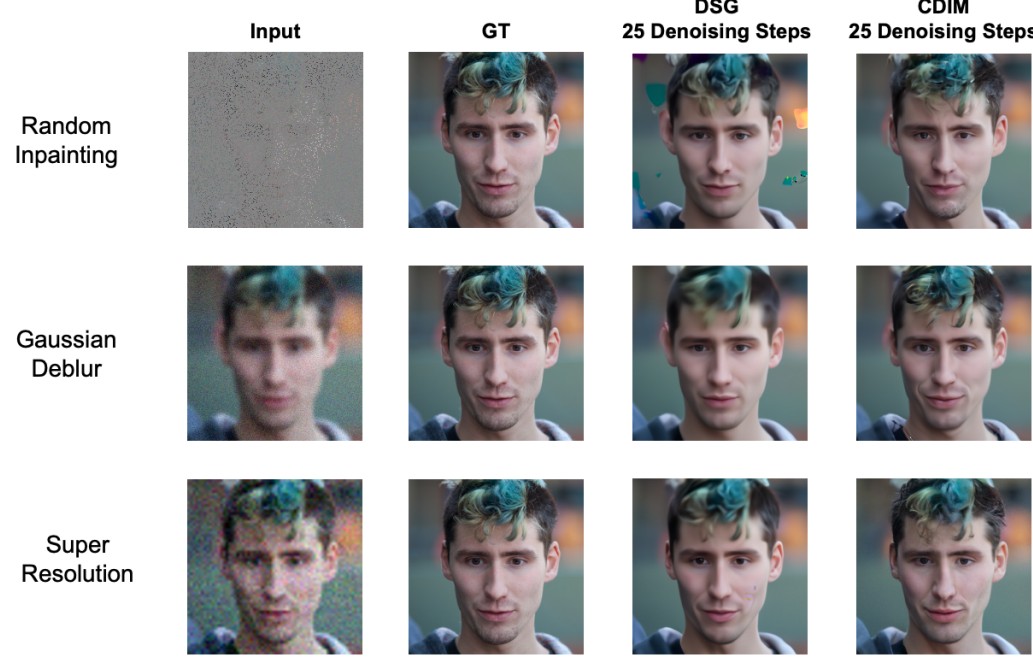

Figure 10: A comparison between CDIM and DSG [28] when both algorithms use 25 DDIM denoising steps. Notice the artifacts in the DSG random inpainting.

## F.2  Comparison with Diff-PIR

Below we report quantitative comparisons against Diff-Pir [25] which uses 100 NFEs. These numbers are reported directly from their paper. Alhtough the LPIPs numbers are comparable or better with Diff-PIR on FFHQ, the FID numbers are noticeably worse and LPIPs is also worse on Imagenet.

Table 6: Comparison between CDIM and Diff-PIR [25] on the FFHQ dataset.

| FFHQ | Super Res | | Gaussian Deblur | | NFEs |
|---|---|---|---|---|---|
| Methods | FID | LPIPS | FID | LPIPS | |
| Ours $T' = 25$ | 33.87 | 0.276 | 34.18 | 0.276 | $\sim$50 |
| Ours $T' = 50$ | 31.54 | 0.269 | 29.68 | 0.252 | $\sim$100 |
| Diff-PIR | 65.77 | 0.260 | 59.65 | 0.236 | 100 |

Table 7: Comparison between CDIM and Diff-PIR [25] on the Imagenet dataset.

| Imagenet | Super Res | | Gaussian Deblur | | NFEs |
|---|---|---|---|---|---|
| Methods | FID | LPIPS | FID | LPIPS | |
| Ours $T' = 25$ | 53.70 | 0.378 | 56.10 | 0.393 | $\sim$50 |
| Ours $T' = 50$ | 47.45 | 0.339 | 38.69 | 0.347 | $\sim$100 |
| Diff-PIR | 106.32 | 0.371 | 93.36 | 0.355 | 100 |

### F.3 Comparison with DAPS

Below we report quantitative comparisons against DAPS-50 [18] which uses 50 NFEs. We run the official code on 100 test images from FFHQ and report the LPIPS loss. These numbers differ from the numbers in the paper because (1) the paper uses $\sigma_y = 0.05$ for images in $[-1, 1]$ while our paper and other baselines have images in $[0, 1]$ which leads to twice as much noise and (2) the paper reports LPIPs with model "alex" while ours and all baselines use "vgg". Both of these cause the reported numbers in the paper to be significantly better, so we run experiments ourselves to ensure a fair comparison.

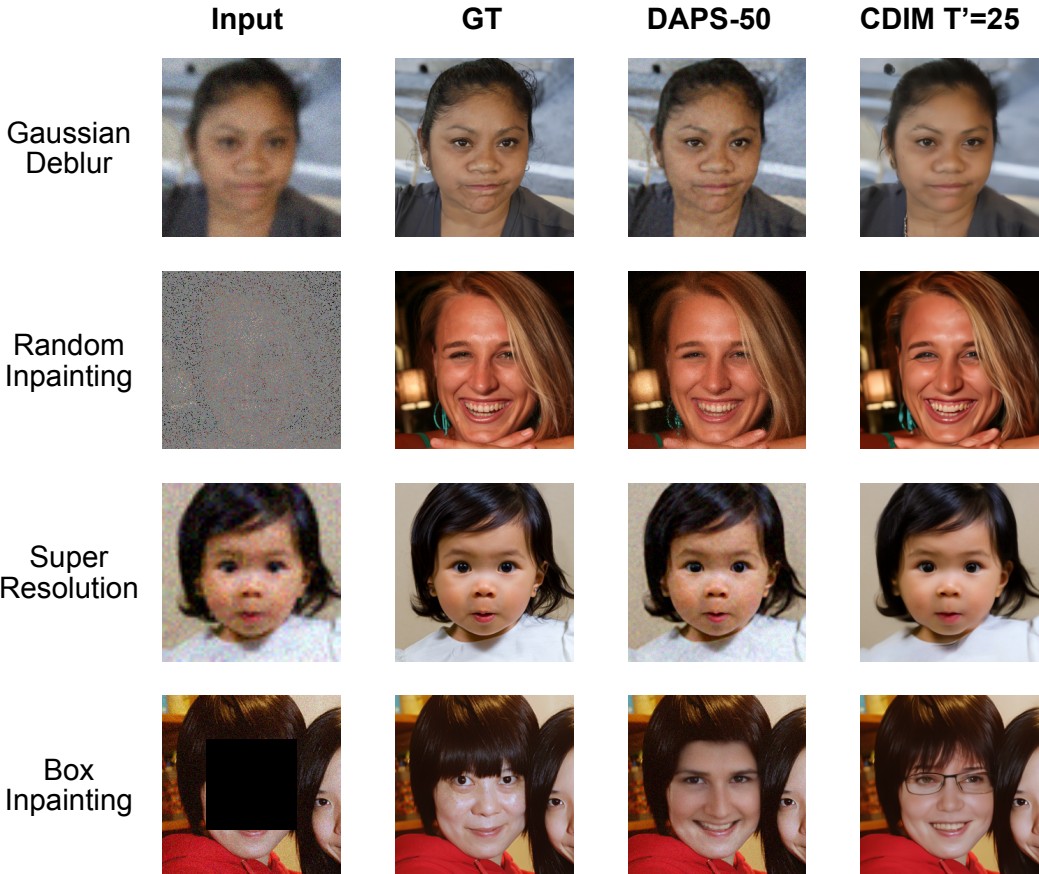

Figure 11: Comparison between CDIM (T'=25) and DAPs-50 where both are capped at 50 NFEs. All of the DAPs-50 results contain noticeable artifacts compared to CDIM.

Table 8: Comparison between CDIM ($T' = 25$) and DAPs-50 on the FFHQ dataset.

| Methods | 4× Super-Resolution | Box Inpainting | Gaussian Blur | Random Inpainting |
|---|---|---|---|---|
| DAPs-50 | 0.358 | 0.244 | 0.339 | 0.306 |
| CDIM ($T' = 25$) | 0.290 | 0.187 | 0.283 | 0.271 |

## G  Extended Results

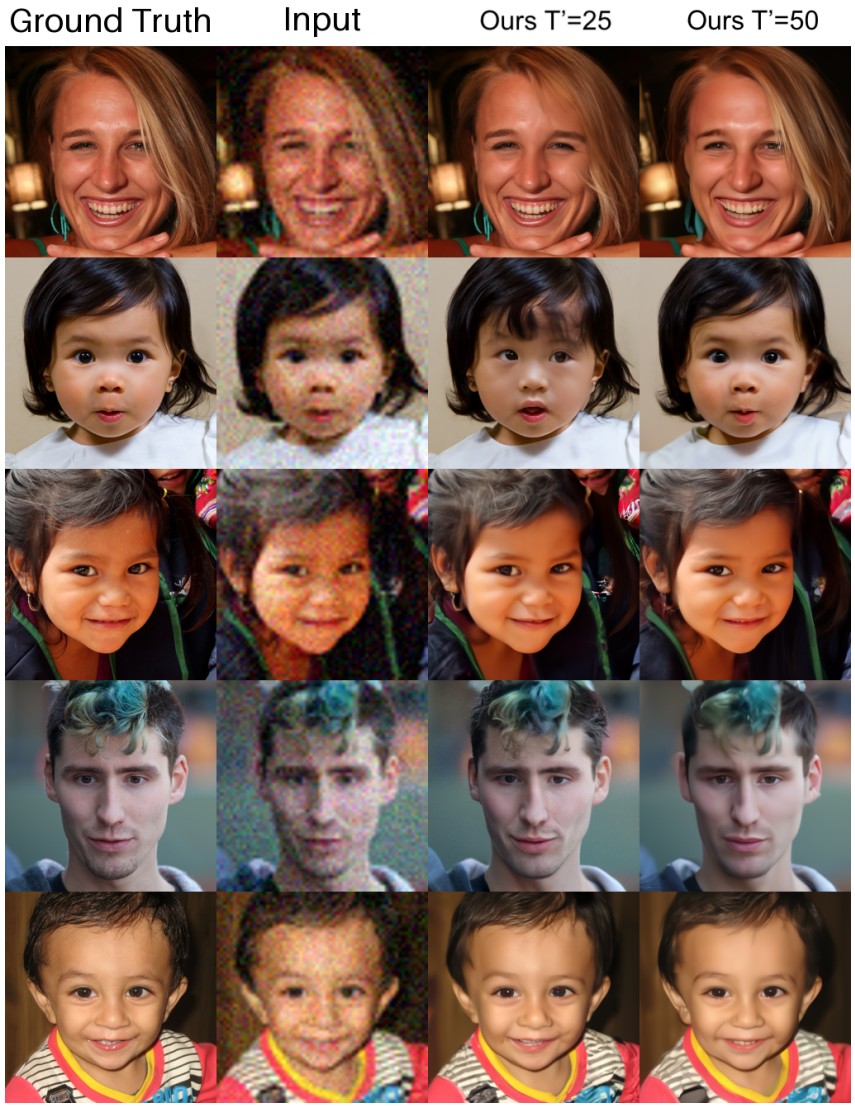

Figure 12: FFHQ Super-resolution extended results

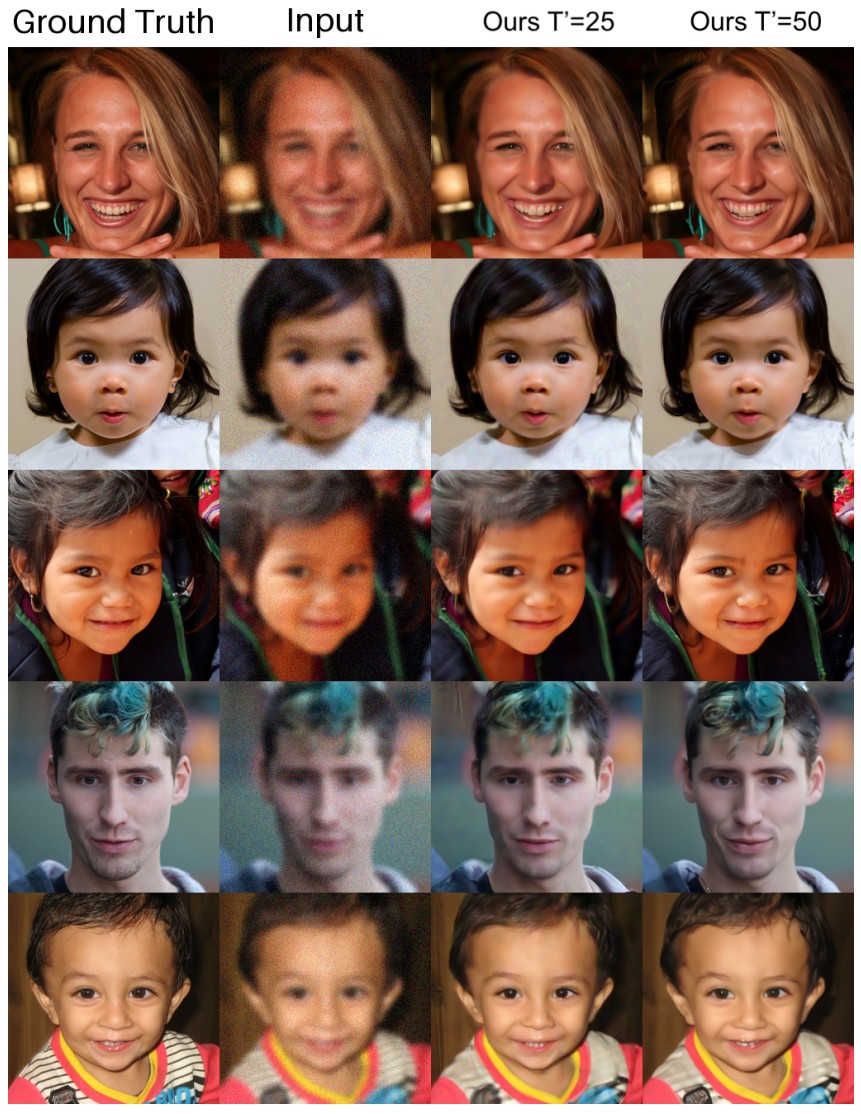

Figure 13: FFHQ Gaussian deblur extended results

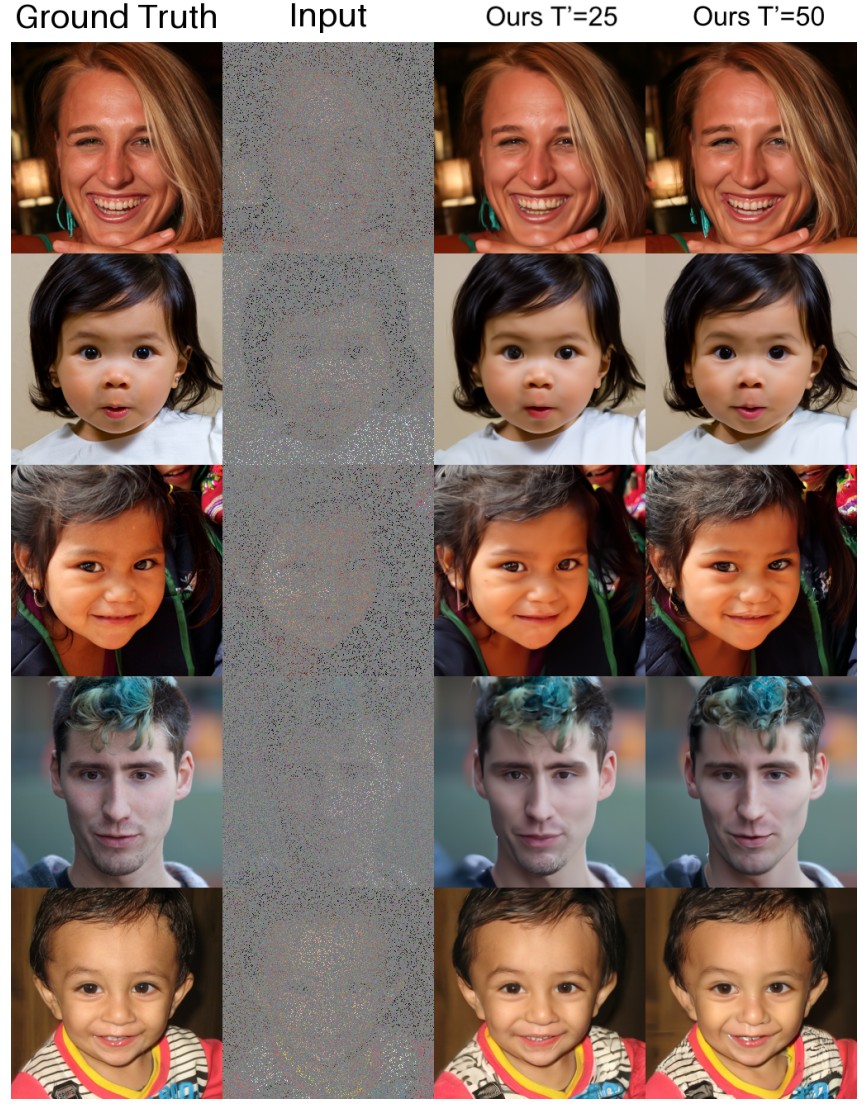

Figure 14: FFHQ random inpainting extended results

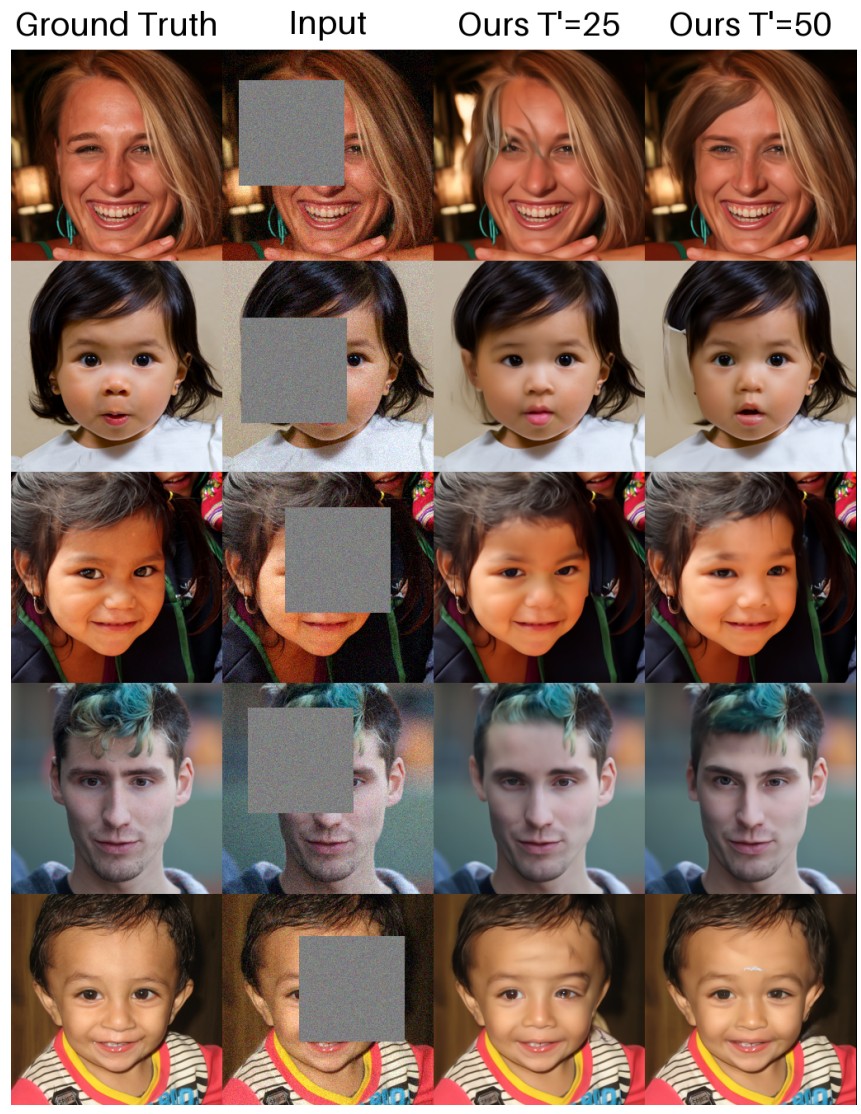

Figure 15: FFHQ box inpainting extended results

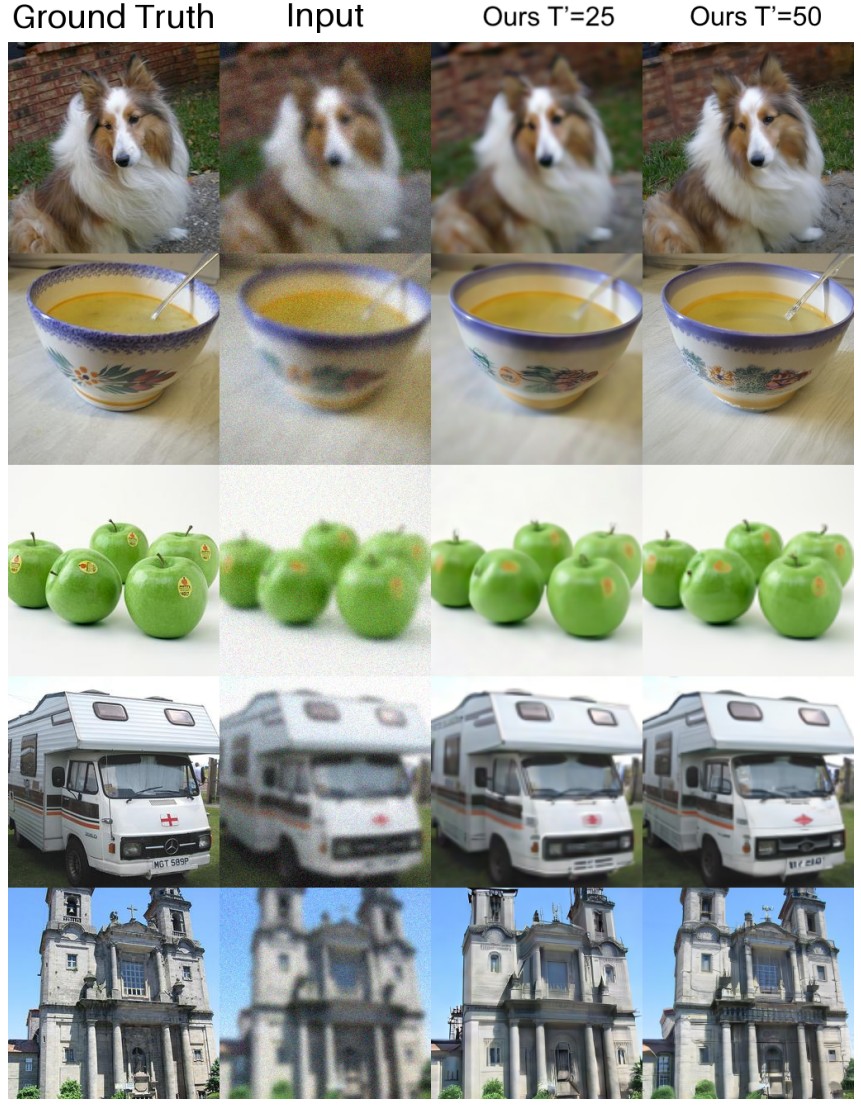

Figure 16: ImageNet Gaussian deblur extended results

Ground Truth    Input    Ours T'=25    Ours T'=50

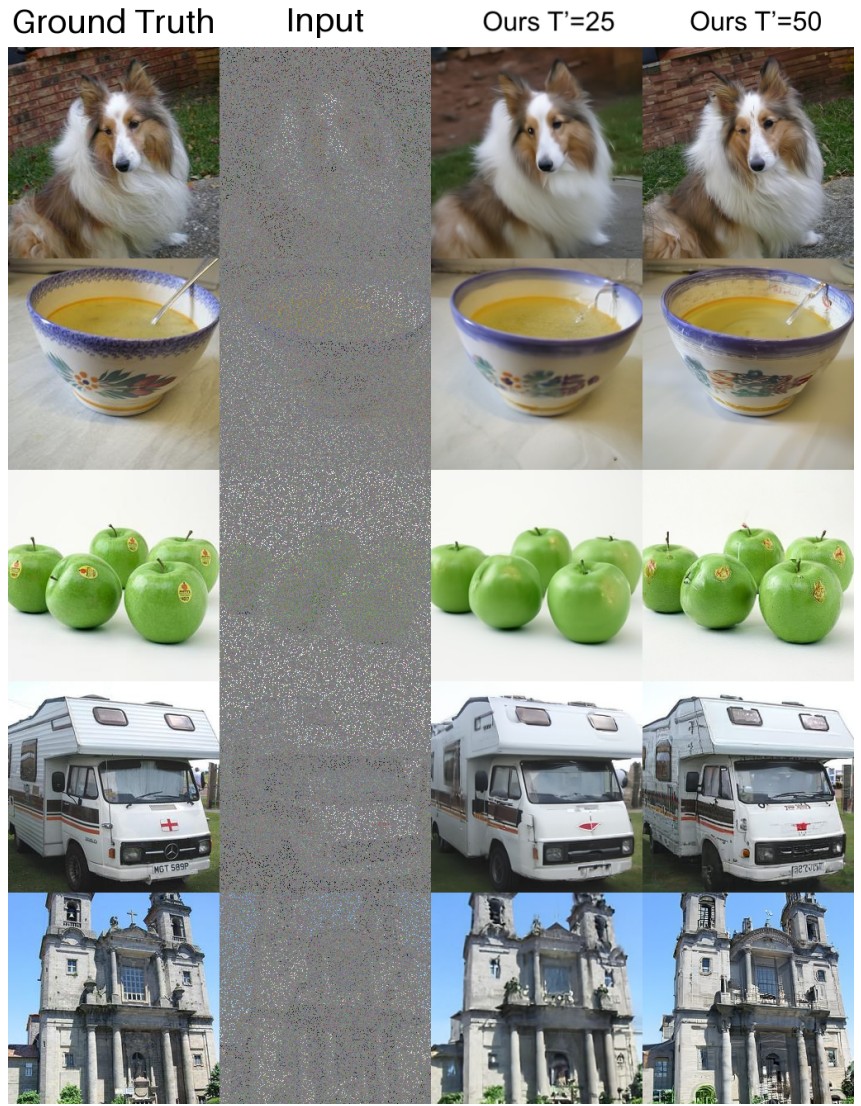

Figure 17: ImageNet random inpainting extended results

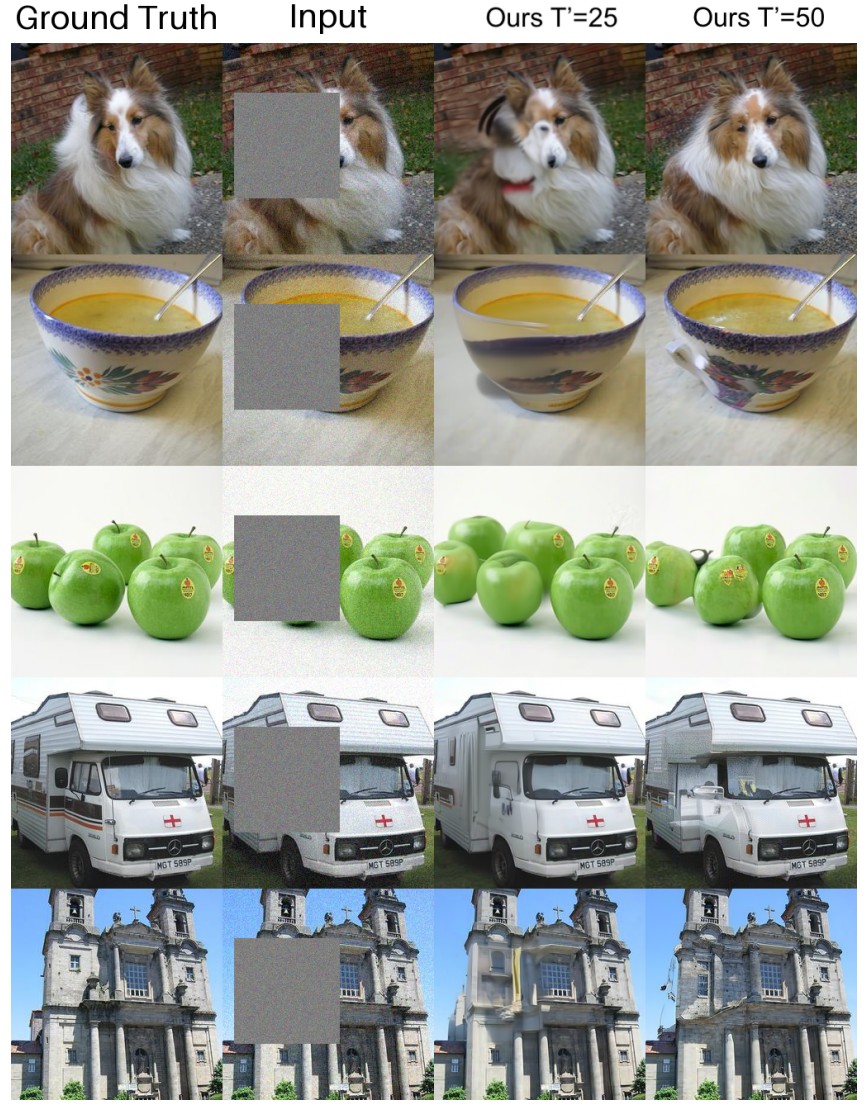

Figure 18: ImageNet box inpainting extended results

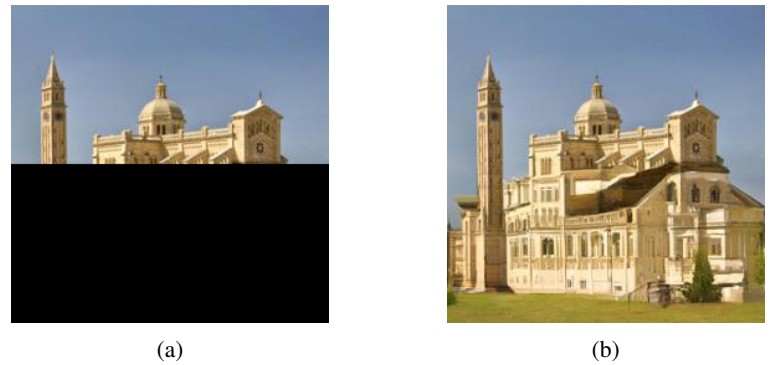

(a)                                (b)

Figure 19: Results on inpainting 50% of an image on LSUN Churches dataset.

