# OpenReview forum: "Linearly Constrained Diffusion Implicit Models"
_NeurIPS.cc/2025/Conference — NeurIPS 2025 poster_

### Official Review · Reviewer_yBWp · 2025-06-21

**Clarity:** 2
**Significance:** 2
**Originality:** 2
**Rating:** 4
**Confidence:** 4

**Summary:**

This work introduces new techniques for solving linear inverse problems with diffusion model priors. The work extends the sample-then-project approach of diffusion posterior sampling with a multi-step projection approach guided by the distribution of the residual under the forward process. The authors show how to define appropriate stopping criteria for the optimization at each step and employ an efficient line search for speed. In their experiments, the authors show that their approach is competitive with other recent approaches to solving linear inverse problems such as inpainting and deblurring as measured by FID and LPIPS, while being efficient in terms of runtime.

**Questions:**

I don't fully understand the paragraph from 145-150. Should the equation on line 147 be $\hat{x}_0$? Could you shed some more light on why is this the best way to define the plausible region?

**Ethical Concerns:**

["NO or VERY MINOR ethics concerns only"]

**Final Justification:**

The method is interesting, but the results could be significantly improved.

**Limitations:**

Yes

**Quality:**

3

**Strengths And Weaknesses:**

Stengths:

The runtime benefits of the proposed approach seem notable and are a significant improvement over some methods.The quality of the solutions are not necessarily state of the art, but appear to be quite good despite the fast sampling time.

The proposed heuristics for step-size and stopping criteria appear to be reasonably well-motivated, with some caveats. The experiments are using difficult real-world datasets and tasks with a wide variety of baselines

The background writing is clear and complete.

Weaknesses:

The novelty is somewhat limited. The key update in equation 12 is the same as DPS and the idea of using multiple projection steps at each sampling time has been discussed in prior work, not cited here. In particular "Constrained Diffusion with Trust Sampling" for multiple step. "Towards Coherent Image Inpainting Using Denoising Diffusion Implicit Models" optimizes a version of equation 11, while combining with other techniques such as multi-step approximation and time-travel. Given the relationship between this method and those, they should be cited and how this method compares to and builds on them should be discussed.

Unlike other extensions of DPS, such as PSLD ("Solving Linear Inverse Problems Provably via
Posterior Sampling with Latent Diffusion Models") this approach doesn't seem to extend to latent diffusion models or general non-linear inverse problems, limiting its usefulness for many state of the art image generation models. The model used for imagenet experiments is somewhat outdated as model for natural images.

More qualitative examples of the results would be useful, particularly for more challenging tasks such as in-painting. I would like to be more confident the results are not cherry-picked.

Some of the writing for the methods section is somewhat confusing. The notation for $\hat{x}$ seems inconsistent, for instance in equations 10 & 11, its not clear if it's meant to approximate $E[x_0 | x_t]$ or $E[x_0 | x_{t-1}]$. It's slightly odd that equation 12 is not directly optimizing eq. 11, though this does end up being the same as DPS.

---

> ### Author Rebuttal · Authors · 2025-07-31
>
> We thank the reviewer for their feedback and comments. We appreciate that the reviewer finds that our method produces good results and that the runtime improvements are notable and significant. We have addressed all the questions and hope the reviewer considers the additional evidence towards a score improvement.
>
> We appreciate the two additional papers that are closely related. We have added these to the related works with a discussion. Specifically, the below is a comparison between our method and the two that are mentioned:
>
> Constrained Diffusion with Trust Sampling: This paper follows a similar motivation by noting that a single or fixed number of optimization steps at each denoising step is sub-optimal. They take multiple optimization steps based on a trust region that considers the variance of $x\_t$ under the forward noising process. In contrast, our “trust region” is based on the analytical distribution of $||Ax\_t - y||^2$ under the forward noising process. Furthermore they use a constant step size while we perform a line search for the optimal step size. As a result, their method requires 1000 NFEs (200 DDIM steps) while ours requires ~50 or ~100 NFEs for high quality results.
>
> Towards Coherent Image Inpainting Using Denoising Diffusion Implicit Models: COPAINT does Bayesian posterior shaping (jointly optimizing revealed and unrevealed regions with approximated likelihoods) whereas CDIM does projection-based constraint enforcement (optimize a simple quadratic residual after each diffusion step). COPAINT is heavier per step but builds coherence gradually; CDIM is simpler and faster, enforcing constraints deterministically at every iteration.
>
> \
> **“This approach doesn't seem to extend to latent diffusion models or general non-linear inverse problems”**
> We note that, like DPS, our method can be extended to non-linear problems but without the theoretical guarantees of exact constraint satisfaction and with using significantly more denoising steps. In this paper, we specifically wanted to focus on exact constraint matching and fast sampling, which are only possible for linear problems. We will add a deeper discussion of nonlinear problems of this in the paper and explain how they can be solved with more steps via a linearized approximation of a non-linear operator $A$, which converges on the constraint more slowly.
>
> \
> **“I don't fully understand the paragraph from 145-150. Should the equation on line 147 be $\hat{x}_0$”**
> Yes! Thanks so much for pointing out this typo. The observed residual on line 147 should indeed by $||A\\hat{x}\_0 - y||^2$. Hopefully this clears up any confusion.
>
> \
> **“Why is this the best way to define the plausible region”**
> We draw from several works (cited on line 149) showing that denoising sampling should mirror forward dynamics as closely as possible. Because the forward noising model adds zero mean Gaussian noise, the conditional mean $\\mathbb{E}\_{\varepsilon_t}\\left[ ||A \\hat{x}\_0 - y||^{2} \\mid y \\right]$  is exactly the most probable (maximum-likelihood) value that this residual can take. Defining the ``plausible'' band as a narrow neighborhood around that mean therefore keeps our reconstructions statistically typical, anything much smaller or larger would be exponentially unlikely under the model and would signal over or under projection.
>
> \
> **“The notation for $\\hat{x}\_0$ seems inconsistent”**
> We appreciate the suggestion on the notation of $E[x\_0 | x\_t]$, which has been mentioned by other reviewers. We have changed the notation $\\hat{x}\_0$ to $\\hat{x}\_0(x\_t)$ throughout the paper to clarify this point

---

> > ### Comment · Reviewer_yBWp · 2025-08-06
> >
> > Thank you to the authors for their rebuttal and my sincere apologies for not having a chance to respond until now. I appreciate the discussion of related methods and the clarification for my questions about the method.
> >
> > With these updates, I will reconsider my score, but I still have strong reservations about the evaluation. Specifically the lack of results on more modern models such as latent diffusion models and the fact that the results on ImageNet seem less promising. In future revisions I would like to see ImageNet results moved to the main text and more qualitative examples in the appendix.

---

> > > ### Author Response · Authors · 2025-08-06
> > >
> > > We thank the reviewer for reconsidering in light of the rebuttal. Based on the feedback, we will add imagenet quantitative examples to the main body and more qualitative examples to the appendix. Note that Figure 6 (the movie pointcloud reconstruction) was done with the imagenet model, showing that it can produce visually compelling results.
> > >
> > > Several other reviewers asked for comparisons with more recent methods. Below we include comparisons against DAPs a very recent (CVPR 2025) and state of the art method.
> > >
> > > DAPs exhibits strong performance, and the results in the main table of DAPs are for DAPs-1k (1000 NFEs). To facilitate a fair comparison, we used the official code and ran the configuration for DAPs-50 (50 NFEs) which has a comparable runtime as CDIM with T’ = 25 denoising steps. Below are the results on 100 images from the FFHQ test set reporting LPIPS (net=”vgg”). Lower is better.
> > >
> > > Random inpainting:
> > > DAPs-50 : 0.306 LPIPS
> > > CDIM (T’=25): 0.271 LPIPS
> > >
> > > Gaussian blur:
> > > DAPs-50: 0.339 LPIPS
> > > CDIM (T’=25): 0.283 LPIPS
> > >
> > > 4x Super Resolution:
> > > DAPs-50: 0.358
> > > CDIM (T’=25): 0.290
> > >
> > > As you can see, CDIM outperforms DAPs when both are restricted to the same fast sampling schedule (even though DAPs-1k exhibits stronger performance with 10x the runtime as CDIM).

---

### Official Review · Reviewer_XQx7 · 2025-06-29

**Clarity:** 3
**Significance:** 3
**Originality:** 3
**Rating:** 4
**Confidence:** 5

**Summary:**

This paper proposes Conditional Diffusion Implicit Models (CDIM), a novel framework for solving linear inverse problems using pre-trained diffusion models. CDIM effectively addresses both noisy and noise-free linear inverse problems. By imposing constraints on the prior diffusion objective, it efficiently solves these problems in terms of both computational time and utility. The paper also incorporates techniques such as Early Stopping and an adaptive learning rate to facilitate more efficient convergence. Experimental results demonstrate that CDIM is fast, powerful, and straightforward to implement with pre-trained diffusion models, requiring no additional modules.

**Questions:**

Please address the weakness of the previous part

**Ethical Concerns:**

["NO or VERY MINOR ethics concerns only"]

**Final Justification:**

I think the author solved my concern and I will raised my score to 4

**Paper Formatting Concerns:**

No paper Formatting Concerns

**Quality:**

3

**Strengths And Weaknesses:**

Strengths:
1. At the heart of this work is a novel integration of the DDIM sampling procedure within an optimization-based framework, ensuring close alignment between the posterior mean and the observed data.
2. The manuscript is clearly written, logically organized, and supported by comprehensive experiments.
3. It also provided a analysis how  CDIM projection strategy both enforces measurement consistency and dramatically cuts down the number of diffusion‐model evaluations.


Weakness:
1. While the proposed method shows potential, its performance still lags behind state-of-the-art approaches such as DAPs [6], and it has not been evaluated on nonlinear inverse problems (e.g., nonlinear deblurring or phase retrieval).

2. The experimental section omits comparisons with several recent diffusion-based sampling algorithms for inverse problems [1–7]. At minimum, the authors should benchmark against leading methods like DAPs [6]

[1] Song, Jiaming, et al. "Pseudoinverse-Guided Diffusion Models for Inverse Problems." ICLR 2023.

[2] Song, Jiaming, et al. "Loss-Guided Diffusion Models for Plug-and-Play Controllable Generation." ICML 2023.

[3] Li, Xiang, et al. "Decoupled Data Consistency with Diffusion Purification for Image Restoration." arXiv preprint arXiv:2403.06054 (2024).

[4] Wu, Zihui, et al. "Principled Probabilistic Imaging using Diffusion Models as Plug-and-Play Priors." arXiv preprint arXiv:2405.18782 (2024).

[5] Xu, Xingyu, et al. "Provably Robust Score-Based Diffusion Posterior Sampling for Plug-and-Play Image Reconstruction." arXiv preprint arXiv:2403.17042 (2024).

[6] Zhang, Bingliang, et al. "Improving Diffusion Inverse Problem Solving with Decoupled Noise Annealing." CVPR 2025.

[7] Dou, Zehao, et al. "Diffusion Posterior Sampling for Linear Inverse Problem Solving: A Filtering Perspective." ICLR 2024.

---

> ### Author Rebuttal · Authors · 2025-07-31
>
> We thank the reviewer for their positive comments and helpful suggestions. We are glad that the reviewer appreciated the novelty of the method and our comprehensive experiments, and also gave us a good score for all areas: quality, clarity, significance, and originality.
>
> We appreciate the additional comparisons suggested. DAPs is a very recent state-of-the-art method that we did not initially compare against, so we report comparison numbers below. These will also be present in the paper revision both in the table and in figure 1.
>
> DAPs exhibits strong performance, and the results in the main table of DAPs are for DAPs-1k (1000 NFEs). To facilitate a fair comparison, we used the official code and ran the configuration for DAPs-50 (50 NFEs) which has a comparable runtime as CDIM with T’ = 25 denoising steps. Below are the results on 100 images from the FFHQ test set reporting LPIPS (net=”vgg”). Lower is better.
>
> Random inpainting:\
> DAPs-50 : 0.306 LPIPS\
> CDIM (T’=25): 0.271 LPIPS
>
> Gaussian blur:\
> DAPs-50: 0.339 LPIPS\
> CDIM (T’=25): 0.283 LPIPS
>
> 4x Super Resolution:\
> DAPs-50: 0.358\
> CDIM (T’=25): 0.290
>
> As you can see, CDIM outperforms DAPs when both are restricted to the same fast sampling schedule (even though DAPs-1k exhibits stronger performance with 10x the runtime as CDIM).
>
> We also have numerous qualitative comparisons that we will include in the revision and would like to share with the reviewer when possible. The DAPs-50 images are slightly noisy due to the accelerated schedule, and the quality improvement is visibly apparent when using CDIM.
>
> We can therefore see that when restricted to a fast sampling environment (50 NFEs or 4 seconds), our method outperforms DAPS, a very recent and state-of-the-art method.
>
> We also thank the reviewer for the other references. We will make sure to include them all in the related work.
>
> Regarding non-linear problems, we can solve them by approximating the stopping criteria $\mu_t(y)$ and $\sigma_t(y)$ via a linearized approximation of the non-linear operator A. Although this method works to comparable quality as baselines when using a large number of denoising steps, we lose both the theoretical guarantees of constraint convergence and the high-quality results during accelerated schedules. This is why we didn't want to focus on nonlinear problems, as we wanted to focus on speed and constraint convergence. Nevertheless we will add this discussion into the paper to show how non-linear problems could be tackled with more denoising steps. (The need for more denoising steps when using nonlinear operators has been discussed in prior works, with DAPs using 4000 NFEs for their latent/non-linear results and only 1000 for linear results.)

---

### Official Review · Reviewer_kQS1 · 2025-07-02

**Clarity:** 3
**Significance:** 2
**Originality:** 2
**Rating:** 4
**Confidence:** 4

**Summary:**

The authors introduce CDIM, a method for solving noisy linear inverse problems using pretrained diffusion models. CDIM modifies DDIM sampling by projecting each step to enforce consistency with linear measurements. It handles Gaussian and Poisson noise, achieves exact recovery in the noiseless case, and performs well on tasks like inpainting and super-resolution. The authors conduct experiments across multiple linear inverse problems and 3D point cloud reprojection.

**Questions:**

Why are variational methods, such as RedDiff, and recent flow-based baselines omitted from the comparisons?

How does the method perform on more challenging settings like 8x SR?

Can the approach be extended to latent diffusion models or nonlinear measurement settings, or is it inherently limited to pixel-space and linear operators?

**Ethical Concerns:**

["NO or VERY MINOR ethics concerns only"]

**Final Justification:**

The authors addressed my main concerns in the rebuttal.

**Limitations:**

The paper does not discuss any limitations, despite the fact that the method cannot be applied to nonlinear forward operators.

**Paper Formatting Concerns:**

No concerns

**Quality:**

3

**Strengths And Weaknesses:**

Strengths:
The paper is clearly written, well-motivated, and presents its core ideas in a structured and accessible manner. The proposed projection-based modification of DDIM sampling is easy to implement and allows for consistent reconstruction under linear constraints.

Weaknesses:
Despite these strengths, the contribution is somewhat limited in novelty. The primary reason for the speedup is the use of DDIM instead of DDPM, which is already well-established. Several strong baselines are missing, including variational approaches [1] and recent flow-based methods that currently define the state of the art. Some of the experiment, such as 4x super-resolution or sparse inpaintin are too simple and could be effectively addressed with classical or non-generative methods, limiting the practical relevance of the comparisons. Lastly, the method is restricted to pixel-space models due to its reliance on linear measurement operators, and cannot be applied in latent diffusion or nonlinear inverse problems.

[1] A Variational Perspective on Solving Inverse Problems with Diffusion Models

---

> ### Author Rebuttal · Authors · 2025-07-31
>
> We thank the reviewer for their comments. We are slightly confused by the reviewer’s summary of the paper, as they mention how our method uses KL divergence minimization, something that is not mentioned once in the paper. KL divergence and the L2 residual minimization that the reviewer mentions are from a non-anonymized arxiv preprint of the paper from several months ago. The version submitted for review has a substantially different method that does not incorporate KL divergence minimization or discuss L2 residual minimization. We would like to make sure that the reviewer has engaged with the method submitted for review, given that the primary weakness stated is the claim that our method is not novel.
>
> \
> **“The primary reason for the speedup is the use of DDIM instead of DDPM”**
>
> We have shown experimental results that simply using DPS with DDIM does not yield good results (see Figure 5). Furthermore, methods like DPMC use DDIM and still require over 200 NFEs to get good results. The speedup comes from our conditional step (DDIM is only for the unconditional steps), which enables fast convergence to the constraint in fewer overall NFEs than previous methods.
>
> \
> **“Why are variational methods, such as RedDiff, and recent flow-based baselines omitted from the comparisons?”**
> Based on feedback from reviewers we have added a comparison with DAPs, a recent state of the art work from CVPR 2025 that outperforms RedDiff.
>
>  To facilitate a fair comparison, we used the official code and ran the configuration for DAPs-50 (50 NFEs) which has a comparable runtime as CDIM with T’ = 25 denoising steps. Below are the results on 100 images from the FFHQ test set reporting LPIPS (net=”vgg”). Lower is better.
>
> Random inpainting:\
> DAPs-50 : 0.306 LPIPS\
> CDIM (T’=25): 0.271 LPIPS
>
> Gaussian blur:\
> DAPs-50: 0.339 LPIPS\
> CDIM (T’=25): 0.283 LPIPS
>
> 4x Super Resolution:\
> DAPs-50: 0.358\
> CDIM (T’=25): 0.290
>
>
> As you can see, CDIM outperforms DAPs when both are restricted to the same fast sampling schedule (even though DAPs-1k exhibits stronger performance with 10x the runtime as CDIM).
>
> We also have numerous qualitative comparisons that we will include in the revision and would like to share with the reviewer. The DAPs-50 images are slightly noisy due to the accelerated schedule, and the quality difference is visibly apparent.
>
> We can therefore see that when restricted to a fast sampling environment (50 NFEs or 4 seconds), our method outperforms DAPS, a very recent and state-of-the-art method.
>
> \
> **“How does the method perform on more challenging settings like 8x SR?”**
>
> Below we include PSNR and LPIPS on 8x superresolution with $sigma_y = 0.05$ Gaussian measurement noise with T’=50 steps (approx 10 second inference) .Given that we cannot share images with the reviewer, we cannot send the qualitative 8x super resolution results, but will include some examples in the revised paper.  Regarding more challenging tasks, we also show several qualitative examples in the paper of 50% inpainting, a significantly harder task than the 25% box inpainting task used in the experiment section.
>
> 8x superres results\
> FFHQ\
> LPIPS: 0.353\
> PSNR: 23.63
>
> ImageNet\
> LPIPS: 0.457\
> PSNR: 19.67
>
> \
> **“Can the approach be extended to latent diffusion models or nonlinear measurement settings, or is it inherently limited to pixel-space and linear operators?”**
>
> Just like DPS, the method can work on latent and non-linear operators, but the theoretical guarantees fail since we rely on linearity of expectation of the Tweedie’s estimate: A(E(x)) = E(A(x)). Therefore the quality improvements in highly accelerated sampling schedules is not as evident for non-linear operators. We have run phase retrieval and non-linear deblurring and plan to include those results in the revision.
>
> \
> **“The paper does not discuss any limitations despite the fact that the method cannot be applied to nonlinear forward operators”**
>
> Note that in the Conclusion we discuss how we do not handle non-linear constraints because of the linearity of expectation.

---

> ### Comment · Reviewer_kQS1 · 2025-08-04
> **Rebuttal**
>
> I thank the reviewers for addressing my concerns and will raise my voting to 4.
>
> Confusion of summary:
> Thank you for pointing this out. The confusion likely arose due to substantial conceptual overlap with the earlier version. I want to assure the authors that I have reviewed the current version of the submission.

---

### Official Review · Reviewer_iTEy · 2025-07-02

**Clarity:** 3
**Significance:** 3
**Originality:** 2
**Rating:** 4
**Confidence:** 3

**Summary:**

The goal of this paper is to solve linear inverse problems using a pretrained generative model. While many existing methods in the literature (e.g., guidance-based or Plug-and-Play approaches) can tackle such problems, none of them explicitly enforce the linear constraint $Ax = y$ in the case of noiseless degradation. However, in certain applications—such as inpainting—it is meaningful to impose a hard constraint, since observed (non-masked) pixels should remain unchanged during reconstruction.

This paper aims to address this limitation by proposing an efficient method for solving inverse problems with hard constraints. The main contributions are:

- An accelerated DDIM-based scheme for the prior denoising step.

- A penalized formulation coupled with gradient descent to enforce constraint satisfaction.

- Practical heuristics for step size selection during optimization.

- Experimental results demonstrating competitive visual quality, faster computation compared to state-of-the-art methods, and better constraint satisfaction.

**Questions:**

First of all, I would like to thank the authors for this paper. I believe the two main questions addressed—hard constraints and acceleration—are relevant for the community. There are some interesting ideas in the paper, but I think the role of each component deserves further clarification.

**1. On the empirical results and ablations**

I appreciated the ablation study against DPS (Fig. 5). Do you have an explanation for why DPS performs poorly when combined with DDIM? Apart from the use of DDIM and the adaptive step size, is there any fundamental difference between your method and DPS?

In the same figure, the claim that “DPS doesn’t match the constraint” is not immediately obvious. You might consider reporting the numerical error on the constraint $Ax = y$ for each method.

It would also be interesting to highlight your method on box inpainting tasks, where other methods may suffer from hallucinations (i.e., inventing content in known regions). This would make the hard constraint enforcement more visible.

**2. On the theoretical justification of constraint satisfaction**

Unfortunately, I remain unconvinced by the theoretical arguments regarding constraint satisfaction:

a. Once the constrained optimization problem is replaced with a penalized one, there is little to no guarantee that the constraints will be exactly satisfied.

b. The justification for the penalized formulation (footline on p.4) is questionable. If I understand correctly, the constraint $A \hat{x_0}(x_{t-1}) = y$ should hold at $t = T$, not $A \hat{x_0}(x_t) = y$. So the argument seems invalid as stated.

c. Referring to Eq. (11) as a “Lagrangian” is incorrect. The formulation lacks dual variables and does not follow the standard Lagrangian structure.

d. I went through the proof in Appendix C and found several steps questionable. For instance:
How do the assumptions in lines 336–337 imply that $x_t$ is bounded?
I disagree with the claim made in line 348.
There seems to be a confusion in the status of $x_t$: sometimes it is defined by the diffusion model, other times it is treated as an optimization variable. In Eq. (37), for instance, $x_t$ is a variable being optimized, so the statement that $|A \hat{x}_0 - x_t| \to 0$ is not meaningful in this context.

e. Have you considered using an Augmented Lagrangian approach? While this may come at a computational cost, it would provide a principled way to handle hard constraints.

**3. Additional Baseline**
It would be useful to include a numerical comparison with Diff-PIR [1], which is a fast PnP method and may serve as a relevant baseline.

**4. Additional suggestions and clarifications**

- I could not fully understand Proposition 1. It seems the notations it relies on are only introduced later, making it hard to follow.

- The Introduction and Related Work sections could benefit from further elaboration. In particular, it would help to provide more details about DPS, which is your main point of comparison.

- To improve readability, I suggest consistently using the notation $\hat{x}_0(x_t)$ to emphasize the dependence on $x_t$ in the reconstructed estimate.

[1] Zhu, Y.; Zhang, K.; Liang, J.; Cao, J.; Wen, B.; Timofte, R. & Van Gool, L. Denoising diffusion models for plug-and-play image restoration, IEEE/CVF Conference on Computer Vision and Pattern Recognition (CVPR) Workshop, 2023, 1219-1229

**Ethical Concerns:**

["NO or VERY MINOR ethics concerns only"]

**Final Justification:**

The authors were cooperative and addressed my questions. The additional numerical results convinced me that the method is valuable, especially given that relatively few papers tackle hard-constrained inverse problems. The authors also acknowledged that the paper is primarily empirical, and that the theoretical arguments are mostly heuristic.

**Limitations:**

yes

**Quality:**

2

**Strengths And Weaknesses:**

**Strengths**

- The paper is well written and relatively easy to follow overall.

- The numerical results are convincing, particularly in terms of computational speed-up compared to methods like DPS.

- The acceleration of the DDIM scheme is an interesting and relevant contribution.


**Weaknesses**

- *Theoretical gaps*: The authors claim that their method satisfies the hard constraints and provide informal arguments in support. However, these arguments appear to be mathematically incorrect or at least imprecise. While the numerical results are more convincing, the theoretical justification remains weak.

- *Lack of focus and clarity on contributions*: The paper seems to pursue two distinct goals:
(1) enforcing hard constraints in inverse problems, and
(2) proposing a fast and visually effective diffusion-based method for inverse problem solving.
The first objective is less convincing, given the theoretical limitations mentioned above. The second is more promising, but the paper lacks a clear analysis of which components of the method (accelerated DDIM, step-size heuristics, gradient descent updates) are most responsible for the performance gains. A more modular ablation or discussion would improve clarity on this point.

---

> ### Author Rebuttal · Authors · 2025-07-31
>
> We thank the reviewer for the detailed and helpful comments. We’re glad they found the results strong and the contribution relevant, and appreciate the careful review of the proofs. Below we address all comments and will fix the issues noted. We hope the author will consider the below for a score improvement.
>
> We would like to start by addressing the main concern regarding the theoretical justification for constraint satisfaction. The reviewer asks about the proof in Appendix C. We note that this proof specifically describes the rate of convergence at which $\lVert A\\hat{x}\_0(x\_{t-1}) - y\rVert^2 \\rightarrow 0$ (which we also demonstrate empirically). However the fact that the constraint can be satisfied follows from the following, much simpler argument: As $t \\rightarrow 0$ the Tweedie's estimate becomes the identity function $\\hat{x}\_0(x_t) \\rightarrow x_t$ and therefore the optimization problem $min\_{x\_{t-1}} ||A\\hat{x}\_0(x\_{t-1}) - y||^2$ becomes the **concave quadratic** $min\_{x\_{t-1}} ||Ax\_{t-1} - y||^2$ . Specifically, at $t-1 = 0$, we can write this out as $x\_0 \gets min\_{x\_0} ||Ax\_0 - y||^2$ and this quadratic can be optimized to arbitrary precision by following the gradients $\\nabla x\_{t-1}$ of our current sample at $t-1 = 0$. We are performing exactly this minimization at $t-1 = 0$ because our stopping criteria is $||Ax\_{t-1} - y||^2 \\leq \\mu_{t-1}(y) \\pm \\sigma\_{t-1}$ where $\\mu\_{t-1}(y) \\rightarrow 0$, $\\sigma\_{t-1} \\rightarrow 0$ as $t-1 \\rightarrow 0$.  By using the optimal step size $\\eta$ via line search, the constraint is met to arbitrary precision (assuming that $y$ is in the range of $A$). This is in contrast to methods like DPS that make a single update to the constraint for each denoising update, and use heuristic step sizes that may not decrease the constraint error to zero. Later in this rebuttal we show quantitatively that the mean absolute pixel error for an observed region in inpainting goes to 0 for our method while it does not for DPS.
>
> \
> **“The paper seems to pursue two distinct goals: (1) enforcing hard constraints and (2) fast diffusion-based inverse problem solving”**
>
> We thank the reviewer for this comment and plan to strengthen our discussion of the connection between these two goals in the revision. These goals are closely related: accelerated unconditional sampling methods like DDIM struggle with constrained sampling because the constraint enforcement steps typically do not converge as fast as the unconditional steps. By proposing a new principled method that quickly decreases the constraints (enforcing the hard constraints even in accelerated sampling), we are able to produce high quality results for fast diffusion-based inverse problem solving. That fact that we guarantee hard constraints is a result of the proposed step size mechanism which is designed for fast posterior sampling by finding the optimal number of steps and step size at each iteration.
>
> \
> **“Do you have an explanation for why DPS performs poorly when combined with DDIM? Apart from the use of DDIM and the adaptive step size, is there any fundamental difference between your method and DPS?”**
>
> DPS takes one constraint step per denoising step with a heuristic step size (footnote 5, p.6 of DPS) and doesn't take into account the additive noise distribution. In contrast, our method proposes a theoretically grounded method for taking optimal steps on the constraint, showing it guarantees constraint recovery and that the optimal step size can be analytically computed without a network pass, using only A’s functional form.
>
> \
> **“DPS doesn’t match the constraint” is not immediately obvious. You might consider reporting the numerical error on the constraint”**
>
> We thank the reviewer for this suggestion. Based on your suggestion we have run the following experiment to report in this rebuttal and for the paper revision. We performed box inpainting with no measurement noise, meaning we should exactly match the constraint, $Ax = y$ on the observed region. We ran both CDIM and DPS with DDIM, each with 50 total NFEs (25 denoising steps). Below we report the per-pixel mean absoluate error $\\frac{1}{m}\\Sigma|Ax\_t - y|$ at the end of select timesteps: This is the measure of how far off the average pixel deviates from ground truth in the known region. (Pixel values between -1 and 1).
>
> T = 960\
> CDIM: 0.72\
> DPS: 0.66
>
> T = 480\
> CDIM: 0.61\
> DPS: 0.62
>
> T = 40\
> CDIM: 0.0058\
> DPS: 0.072
>
> T = 0\
> CDIM: 0.00045 (In theory would be 0 but we have $\\overline{\\alpha}\_0 = 0.9999$ not $1$ for numerical stability)\
> DPS: 0.069
>
>
> CDIM matches the constraint to numerical precision, while DPS differs from observed pixels by an average of almost 7%. Any attempt to make DPS more closely match the constraint, for example increasing the gradient step size, results in divergence for accelerated sampling schedules.
>
>
> \
> **“It would also be interesting to highlight your method on box inpainting task”**
>
> We include several examples in the paper of a 50% inpainting task (harder than box inpainting) where we exactly match the constraint. As an example, consider Figure 3 where sub-panel b where we can force the observed region to match the noisy pixels exactly even when they are out of distribution.
>
> \
> **“a. Once the constrained optimization problem is replaced with a penalized one, there is little to no guarantee that the constraints will be exactly satisfied.”**
>
> We note that as $t \\rightarrow 0$, the penalized optimization turns into the hard constraint problem given by equation (10). This is because our stopping criteria $\\sigma$ goes to 0 as $t \\rightarrow 0$, so there is no early stopping. We continue running optimization steps on $||A\\hat{x}\_0(x\_{t-1}) - y||^2$ until we reach the desired level of convergence. Because this is concave quadratic, it can always be optimized to sufficiently meet the hard constraint. Therefore we have guarantee of exact constraint satisfaction at $t=0$ (and only at $t=0$ because higher noise levels do have the early stopping criteria).
>
> \
> **“b. The justification for the penalized formulation (footline on p.4) is questionable. If I understand correctly, the constraint $A\\hat{x}\_0(x\_{t-1})$ should hold at t = T. So the argument seems invalid as stated.”**\
> We thank the reviewer for pointing out this subtle fact. It is true that the constraint in question is $\\hat{x}\_0(x\_{t-1})$, so even at the first optimization step, we are computing the Tweedie’s estimate $E[x\_0 | x\_{T-1}]$ and not $E[x\_0 | x\_{T}]$. However, the intuition holds that even at $t=T-1$, $x\_t$ is largely uninformative, so finding $x\_t$ where $A\\hat{x}\_0(x\_{t-1})=y$ is effectively impossible. The tweedie’s estimate is the expectation over all denoising trajectories, which would still be a blurry image at high noise values, even if say we set $x\_{T-1}$ to our GT image.
>
> \
> **“c. Referring to Eq. (11) as a Lagrangian is incorrect.”**
>
> We thank the reviewer for pointing out this correction. We will change the description of the equation.
>
> \
> **“d. Proof in Appendix C”**\
> We thank the reviewer for the careful reading and agree that Appendix C requires clearer exposition.
>
>
> *Boundedness of $x_t$*\
> Since the forward process is a convex combination of a bounded datum $x_0$ where $||x\_0|| \\le R$ and Gaussian noise with coefficient $\\sqrt{\\beta_t}\\le 1$, the resulting sample is almost surely bounded: $||x\_t|| \\le R+O(\\sqrt{\\beta_t})$. We will add this explicit bound.
>
> *Role of $x\_t$ versus $\\hat x\_0(x\_t)$*\
> Throughout Appendix C, $x\_t$ is the optimization variable selected by the projection step, while $\\hat x\_0(x\_t)$ is its Tweedie estimate. Equation 37 is saying that the Tweedie’s estimate of our current optimization iterate converges to the identity function. We will make this distinction explicit to avoid confusion. We don't intend to say that $||A\\hat{x}\_0 - x\_t|| \\rightarrow 0$ but that $||\\hat{x}\_0(x\_t) - x\_t|| \\rightarrow 0$, meaning that optimizing the tweedie estimate of the iterate is the same as optimizing the iterate itself at low noise values. We also show this empirically.
>
> *Convexity of the objective as $t\\to 0$*\
> Tweedie convergence gives $||\\hat x\_0(x\_t) - x\_t|| = O(\\beta\_t)$; expanding Eq.~(37) shows that $||A\\hat x\_0(x\_t) - y||^2 = ||A x\_t - y||^2 + O(\\beta\_t)$. Hence, for sufficiently small $t$, the landscape is an arbitrarily small perturbation of the convex quadratic $||A x_t - y||^2$, so standard gradient descent reaches any desired accuracy.
> These clarifications will be incorporated, but they leave the main conclusion unchanged: as $t-1\to 0$, gradient descent on $||A\hat x_0(x_{t-1}) - y||^2$ can satisfy the constraint to arbitrary precision.
>
> **“It would be useful to include a numerical comparison with Diff-PIR [1]”**\
> We thank the reviewer for pointing out Diff-PIR. Below are some direct comparisons reporting LPIPS (lower is better), which are also added to the paper
>
> 4x Super Resolution Imagenet:\
> Diff-Pir (100 NFEs): 0.371 LPIPS\
> Ours (100 NFEs): 0.339 LPIPS
>
> Gaussian Deblur Imagenet:\
> Diff-Pir (100 NFEs): 0.355 LPIPS\
> Ours (100 NFEs): 0.347 LPIPS
>
> With the same number of NFEs, our method performs better.
>
> \
> **“I couldn't fully understand Proposition 1...relies on notation introduced later,”**
>
> Thank you for pointing this out, indeed the notation was introduced after the proposition making it hard to follow. The key idea of Proposition 1 is that we can choose our optimal step size based on the value of $||Ax\_t - y||$ (calculable via line search without a network pass) and that this also guarantees that our true target, $||A\\hat{x}\_0 - y||$ decreases towards its desired value by a commensurate amount.
>
> \
> **“To improve readability, I suggest consistently using the notation $\\hat{x}\_0(x\_t)$”**\
> We thank the reviewer for this suggestion. It has also been suggested by other reviewers and we will change the notation in the revised paper.

---

> ### Comment · Reviewer_iTEy · 2025-08-01
>
> I thank the authors for addressing my questions.
>
> - I appreciate the 2 news experiments, on the constraint satisfaction and comparison to DiffPir. Thanks for that.
>
> - Thanks for pointing to Figure 3, I had missed that one, apologies.
>
> - I appreciate the efforts put into trying to clarify the mathematical arguments in the proof that the constraint is satisfied at time $t = 0$, but I have to say I am still not convinced. The arguments are not rigorous; in particular, saying that it becomes a concave quadratic (did you mean convex?) as $t \to 0$ is incorrect.
> First of all, I find the current notation $x_t \to \lVert A\hat{x}_0(x_t) - y \rVert^2$ not very easy to manipulate from an optimization perspective. Therefore, I will rather denote $f_t(x) = \lVert A \hat{x}_0^t(x) - y \rVert^2$,
> where I included a dependency of $\hat{x}_0$ on $t$, and changed the name of the global variable on which you are optimizing to $x \in \mathbb{R}^d$.
> Then $f_t$ converges pointwise to $f(x) = \lVert Ax - y \rVert^2$, but this does not imply that $f_t$ is convex for some $t$ close to 0.
> I kindly ask the authors not to claim they provide a mathematical proof; to me, this is a heuristic (and heuristics are fine), and it should be clearly stated as such.

---

> > ### Author Response · Authors · 2025-08-02
> >
> > Thank you for engaging with our response.
> >
> > * Regarding notation, we are happy to revise our notation to use $x$ as the optimization variable and work with the time-dependent function $\hat x^t_0(x)$.
> >
> > * Regarding pointwise convergence and convexity as $t \to 0$: your point is well-taken. We will revise the language around line 132 to make clear that $\lVert Ax - y \rVert$ is only convex at $t = 0$ exactly. The fact that we can optimize this term well and remain within the plausible region away from $t = 0$ is a fortunate empirical observation, supported by Figure 7 and the comparative measurement of constraint satisfaction for CDIM, DPS and other algorithms that we provided in our response to your previous comments.
> >
> > * Taking the point about mathematical proof more broadly: we will soften any claims of proof elsewhere in the paper. The intent of our paper is to make an algorithmic contribution with supporting empirical evidence and mathematical intuitions, not to make formal claims about optimizations through neural networks.
> >
> > We are excited about this CDIM method, and you seem to agree that the method is interesting and empirical results are convincing! Does the addition of the provided empirical evidence of constraint satisfaction, along with a softening of language that suggests mathematical proof satisfy your concerns about our paper?

---

> > > ### Comment · Reviewer_iTEy · 2025-08-03
> > >
> > > Thank you for agreeing that the proofs provided are closer to heuristics.
> > > I appreciate your suggestion to include both empirical results and to soften the language (e.g., replacing "proof" with "intuition"). If you accept this change, I would be willing to raise my score to 4.

---

> > > > ### Author Response · Authors · 2025-08-04
> > > >
> > > > Yes, we will include the empirical results on convergence and constraint satisfaction in the main paper and make it clear that our calculations provide intuition for these behaviors, not proof. We will also revise our notation as mentioned in the previous discussion. Thank you again for this feedback!

---

### Official Review · Reviewer_RQq2 · 2025-07-03

**Clarity:** 3
**Significance:** 3
**Originality:** 3
**Rating:** 5
**Confidence:** 3

**Summary:**

The paper introduces CDIM, a method for solving linear inverse problems through adaptive projection sampling (using a pretrained diffusion model). Authors show that under the forward noising process, the measurement residual follows a chi-square distribution, with tractable mean and variance. Using this, CDIM chooses projection step sizes and stopping criteria adaptively and projects $x_t$ just enough to keep residual within one standard deviation of its expected value at each step.
In the noiseless setting, this adaptive method guarantees exact constraint satisfaction and bounds the final reconstruction error with high probability.

Their empirical results show that CDIM achieves image quality on par with SoTA methods like DPS with 50x fewer model evaluations across multiple tasks on FFHQ and ImageNet, and some samples new applications like photo time travel and 3D point-cloud reconstruction.

**Questions:**

1. Would it be possible to extend this method for latent-based models?
3. How sensitive are final results to the choice of "±1 σ" as the stopping criterion? What if you choose a dynamic tolerance (e.g. varying with t) improve performance?
2. Section D of appendix looks incomplete. Subsection D.1 has typos and inconsistency in notation (like "64 ∝ 64" and "showcase a a 50% inpainting"). Subsection D.4 should be referring to table 3?

**Ethical Concerns:**

["NO or VERY MINOR ethics concerns only"]

**Final Justification:**

Authors have addressed my concerns and I continue to support acceptance of this work.

**Limitations:**

Yes

**Quality:**

3

**Strengths And Weaknesses:**

Strengths:
1. The paper proposes a novel method for using pretrained diffusion models to solve inverse problems that uses an adaptive projection mechanism grounded in the residual distribution.
2. The method achieves the same level of measurement fidelity with an order of magnitude fewer denoiser calls.
3. Empirical results are provided on FFHQ and ImageNet datasets which are standard sets for these tasks.

Weakness:
1. In a couple of cases, CDIM shows perofrmance that is close, but not exceeding the best slower baseline. For example, DPS outperformed  on random inpainting and FPS-SMC outperformed CDIM on super resolution FID.

---

> ### Author Rebuttal · Authors · 2025-07-31
>
> We would like to thank the reviewer for their positive feedback and thoughtful comments. We are glad that they appreciate the novelty and strength of the results. Below we address the specific concerns/questions raised. We plan to upload a revision with all these specific fixes.
>
> **“In a couple of cases, CDIM shows performance that is close, but not exceeding the best slower baseline.”**
>
> We acknowledge that this is true, however the baselines mentioned use far more compute than us. When restricting those methods to a same amount of compute (such as the DPS + DDIM image in figure 5), our method performs significantly better.
>
> **“Would it be possible to extend this method for latent-based models”**
>
> We note that, like DPS, our method can be extended to latent-based models with some amount of success. However, because Tweedie’s formula for the expected posterior mean does not pass through a non-linear function, $A(E[\hat{x}_0]) \neq E[A(\hat{x}_0)]$, we cannot get the same theoretical guarantees on convergence of the constraint, or the same speedup. In addition, other heuristics may be required to get satisfactory convergence in latent models, which is why we have left this for a follow up work.
>
> **“How sensitive are final results to the choice of $\pm \sigma$ as the stopping criterion? What if you choose a dynamic tolerance (e.g. varying with t) improve performance?”**
>
> In Figure 4 we have shown qualitative results when different stopping criteria than $\pm 1 \sigma$ are used.  Additionally, in Table 2 we have given quantitative results both on the resulting runtime and the generation quality for a chosen task when using these different stopping criteria. Although it would be possible to choose a dynamic tolerance, we note that the standard deviation $\sigma$ is itself highly informative on the plausible region, decreasing dramatically as t->0, so a constant factor is sufficient.
>
> **“Section D of appendix looks incomplete.. typos and inconsistencies”**
>
> We thank the reviewer for pointing this out. We will address the typos and inconsistency in the revised version. Indeed subsection D.4 should be referring to table 3 and 4, and we will revise the inconsistencies in experimental notation.

---

> > ### Comment · Reviewer_RQq2 · 2025-08-06
> >
> > I appreciate authors' efforts in addressing the comments and remain in support of accepting this work.

---

### Decision · Program_Chairs · 2025-09-17

**Decision:**

Accept (poster)

**Comment:**

All reviewers have positive comments about this work, appreciating that the method offers substantial computational advantages over (strong) baseline methods for inference with pre-trained diffusion models, and is supported by some theoretical analysis of the diffusion process.  In the reviews and discussions, and number of points were raised relative to the clarity/details of the technical development, as well as the experimental results.  Please take care to thoroughly address these points in your final manuscript.